# Inherited defects of piRNA biogenesis cause transposon de-repression, impaired spermatogenesis, and human male infertility

Birgit Stallmeyer[1], Clara Bühlmann[1], Rytis Stakaitis [2,3], Ann-Kristin Dicke [1], Farah Ghieh[1], Luisa Meier[1], Ansgar Zoch[4,5], David MacKenzie MacLeod[4,5], Johanna Steingröver [1], Özlem Okutman[6,7], Daniela Fietz [8], Adrian Pilatz [9], Antoni Riera-Escamilla [10], Miguel J. Xavier [11], Christian Ruckert[12], Sara Di Persio[13], Nina Neuhaus [13], Ali Sami Gurbuz[14], Ahmet Şalvarci[15], Nicolas Le May[6], Kevin McEleny[16], Corinna Friedrich[1], Godfried van der Heijden [17], Margot J. Wyrwoll[1,4], Sabine Kliesch [13], Joris A. Veltman [11], Csilla Krausz[10,18], Stéphane Viville [6,19], Donald F. Conrad [2], Dónal O'Carroll [4,5] & Frank Tüttelmann [1] ✉

piRNAs are crucial for transposon silencing, germ cell maturation, and fertility in male mice. Here, we report on the genetic landscape of piRNA dysfunction in humans and present 39 infertile men carrying biallelic variants in 14 different piRNA pathway genes, including *PIWIL1, GTSF1, GPAT2, MAEL, TDRD1*, and *DDX4*. In some affected men, the testicular phenotypes differ from those of the respective knockout mice and range from complete germ cell loss to the production of a few morphologically abnormal sperm. A reduced number of pachytene piRNAs was detected in the testicular tissue of variant carriers, demonstrating impaired piRNA biogenesis. Furthermore, LINE1 expression in spermatogonia links impaired piRNA biogenesis to transposon de-silencing and serves to classify variants as functionally relevant. These results establish the disrupted piRNA pathway as a major cause of human spermatogenic failure and provide insights into transposon silencing in human male germ cells.

PIWI-interacting RNAs (piRNAs) represent a specific type of regulatory, single-stranded small non-coding RNAs preferentially expressed in germ cells. They are required for transposon silencing, thus safeguarding genome integrity in the fetal and adult mammalian testis, and sculpting the post-meiotic transcriptome[1,2]. In contrast to mice, in which disrupted piRNA biogenesis has been tightly linked to male-specific infertility, the role of the piRNA pathway in spermatogenic failure in men remains largely unexplored.

piRNAs bind to a subclade of Argonaute proteins known as PIWI proteins (derived from P-element-induced-wimpy testis)[2]. Based on their temporal expression in mice, piRNAs are classified into three distinct main categories: fetal, pre-pachytene, and pachytene piRNAs[1]. Fetal piRNAs, which are expressed in prospermatogonia, are loaded into both PIWIL2 and PIWIL4, while pre-pachytene piRNAs, which are already expressed in early spermatogenic cells and are present up to the meiotic pachytene stage, are mainly associated with PIWIL2. Finally, pachytene piRNAs bound by PIWIL1 and PIWIL2 are abundant from the pachytene stage of meiosis until the postmeiotic elongated spermatid stage and account for more than 90% of piRNAs in the adult testis[2-4].

Pachytene piRNAs originate predominantly from non-repetitive, intergenic regions, called pachytene piRNA clusters, and contain only few transposable element (TE) sequences[3,5]. They regulate gene expression by inducing post-transcriptional mRNA degradation[6] or by activating translation at the post-meiotic stages of spermatogenesis[7]. On the contrary, fetal and pre-pachytene piRNAs are enriched in

TE-targeting sequences and essential for their post-transcriptional silencing through the piRNA-induced silencing complex (piRISC) mediated slicer activity[8–10]. In addition, fetal piRNAs are required for de novo transposon methylation[11,12].

piRNA biogenesis can be differentiated into two pathways involving not only PIWI proteins but also Tudor domain-containing proteins (TDRDs) acting as scaffolds, along with several enzymes involved in pre-piRNA trimming and maturation[13] (Supplementary Fig. 1a, b). The biogenesis of pachytene piRNAs is restricted to the primary pathway, in which long piRNA precursors are transferred from the nucleus to the cytoplasm and accumulate in perinuclear structures called nuages. Here, mature piRNAs are produced through cleavage and processing of piRNA precursors. This cleavage is independent of piRISC activity and is initiated by the endonuclease PLD6, which establishes the 5′-ends of pre-piRNAs[14,15]. In contrast, in the fetal testis, complementary long piRNA precursors are mainly cleaved by PIWIL2-bound piRISC complexes. Here, the massive amplification of TE-derived fetal piRNAs is established in the secondary pathway (ping-pong cycle)[2].

Knockout mice for more than twenty genes related to the piRNA-pathway have been analyzed[13]. These mice are concordantly affected by male-specific infertility, small testes, and germ cell maturation arrest at meiosis or early haploid cell stages. Furthermore, a substantial reduction in the amount of piRNAs and consequent de-repression of TEs in the fetal and/or adult testis was observed in several mouse models[16–19].

In men, biallelic variants in several piRNA-related genes have recently been reported to cause infertility due to spermatogenic failure leading to non-obstructive azoospermia (NOA) or cryptozoospermia, i.e., no or very few sperm in the ejaculate[20–23]. However, only biallelic variants in *PNLDC1*, *FKBP6*, and *TDRD9* have as yet been functionally linked to reduced levels of germ cell-derived piRNAs and, thus, impaired piRNA biogenesis[23–25].

Here, we shed light on the impact of disrupted piRNA biogenesis on human spermatogenesis by presenting 39 infertile men carrying rare, biallelic, putative pathogenic variants in 14 different genes encoding proteins of the piRNA pathway. Interestingly, the observed testicular phenotypes repeatedly differ from those of the respective knockout mice. Furthermore, we show that the dysfunction of piRNA pathway proteins in the human adult testis not only leads to a reduced amount of pachytene piRNAs but is also associated with a gene-specific increase of transposon expression in spermatogonia. These analyses can serve as readout for the functional relevance, i.e., pathogenicity, especially of the identified missense variants.

## Results
### Genes of the piRNA pathway are frequently mutated in infertile men

To elucidate protein networks or biological pathways contributing to impaired spermatogenesis, we queried exome/genome data of >2000 infertile men from the Male Reproductive Genomics (MERGE) study[26] for rare homozygous loss-of-function (LoF) variants in genes preferentially expressed in the human testis. On the 61 identified genes, we performed a Gene Ontology (GO)-based two-tiered hierarchical clustering of significantly enriched biological processes that showed a striking enrichment of categories associated with "piRNA processing" (Fig. 1a, Supplementary Fig. 2). Further analysis revealed that piRNA pathway genes also contributed to the most significantly enriched individual processes (Fig. 1b). Next, we screened the MERGE data from 2127 infertile men with azoo-, cryptozoo-, extreme or severe oligozoospermia (<2/<10 million total sperm count; Online methods) for biallelic, high-impact variants (minor allele frequency [MAF] in gnomAD < 0.01; LoF or missense variants with CADD score ≥15) in 24 human orthologues of murine genes associated with piRNA biogenesis (Fig. 1c; Supplementary Table 1) and identified 31 men carrying variants fulfilling the selection criteria (Table 1).

Of these, 27 patients carried homozygous variants (11 LoF and 16 missense with CADD score ≥20) and four patients carried confirmed compound heterozygous variants. In total, these affected 14 different genes: *DDX4*, *FKBP6*, *GPAT2*, *GTSF1*, *HENMT1*, *PIWIL1*, *PIWIL2*, *PLD6*, *PNLDC1*, *MAEL*, *MOV10L1*, *TDRD1*, *TDRD9*, and *TDRD12* (Table 1, Fig. 1c, Supplementary Data 1, also including reference transcripts). The three *FKBP6* variant carriers and also one of the *TDRD12* variant carriers have been described previously[23,25]. Detailed analysis of the exomes from each affected men did not reveal any other variants with a higher probability for causing the disease. In two cases, chromosomal translocations were identified (Supplementary Table 2) and it cannot be excluded that they at least partially contribute to the patient's phenotype. Of the 31 affected men, 19 were azoospermic, nine were cryptozoospermic, and four had extreme oligozoospermia. Twenty-three patients underwent a testicular biopsy with the aim of sperm extraction (TESE), which was negative in 22 men, i.e., no sperm could be obtained. The analyses of testicular tissue revealed phenotypes ranging from complete absence of germ cells (Sertoli cell-only, SCO, $N = 7$), the presence of spermatocytes (meiotic arrest, MeiA, $N = 6$), round spermatids (RsA, $N = 7$) or elongated spermatids (ES+, $N = 3$) as the most advanced germ cells (Supplementary Fig. 3, Supplementary Table 2).

In addition, screening of exome data from three independent cohorts of infertile men identified eight additional patients with homozygous high-impact variants (two LoF and six missense) in *GPAT2*, *PIWIL2*, *MOV10L1*, and *TDRD12* (Table 1, Supplementary Data 1), bringing the total number of variant carriers to 39. In six of the additional patients, the testicular phenotypes of SCO or spermatogenic arrest confirmed the clinically suspected non-obstructive azoospermia (Supplementary Table 2). In summary, 38 distinct variants in piRNA genes were identified among 39 infertile men. Of these variants, 12 were absent from gnomAD (version v2.1.1) and 18 were extremely rare (MAF ≤ 0.0001) (Supplementary Data 1).

### Variants in genes encoding components of the piRISC complex

Pachytene piRNAs have been proposed to direct PIWIL1 and PIWIL2 to cleave specific mRNAs and thereby regulate gene expression[27]. The slicing activity of this piRNA-induced silencing complex (piRISC) requires GTSF1 as an auxiliary factor[28]. We identified four azoospermic men with biallelic variants in genes encoding proteins essential for piRISC activity (Table 1). In *PIWIL1*, a homozygous stop-gain variant c.688C>T p.(Arg230*) localizing within the PAZ domain (Fig. 2a, Supplementary Fig. 4a) was identified. The variant carrier M2006 exhibited CREM-positive, haploid, round spermatids as the most advanced germ cells in the seminiferous tubules (Fig. 2b) and TESE was negative (Supplementary Table 2). Further staining demonstrated the absence of PIWIL1 in testicular germ cells, which is expressed in spermatocytes and haploid germ cells in a control subject with normal spermatogenesis (Fig. 2c).

Furthermore, two azoospermic men carried two different homozygous missense variants, in *PIWIL2*, c.1697G>A p.(Arg566His) in M2949 and c.839A>C p.(Tyr280Ser) in TP32, affecting amino acids conserved up to zebrafish (Supplementary Fig. 4b). Arg566 is predicted to be a surface-accessible residue found within the linker 2 (L2) domain of PIWIL2[29] that bridges the PAZ and MID domains (Fig. 2a). Tyr280 is located within the structured N-terminal region of PIWIL2, which has been suggested to stabilize piRNA-target duplex conformations[30]. Finally, two men with meiotic arrest were carriers of homozygous, high-impact variants in *GTSF1* (Fig. 2a, Supplementary Fig. 4c). The frameshift variant c.221_222del p.(Arg74Lysfs*4) was identified in M2243 and predicted to result in nonsense-mediated decay (NMD), leading to abolished GTSF1 expression in the patient's spermatocytes (Fig. 2c). The missense variant c.97C>A p.(His33Asn) identified in M2043 is located in a predicted α−helical protein domain

**a** Study approach

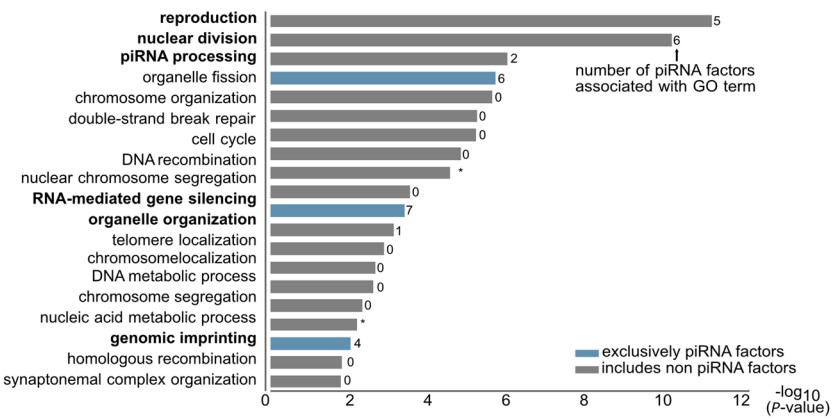

**b** Individual Gene Ontology terms

**c** piRNA biogenesis and piRISC activity

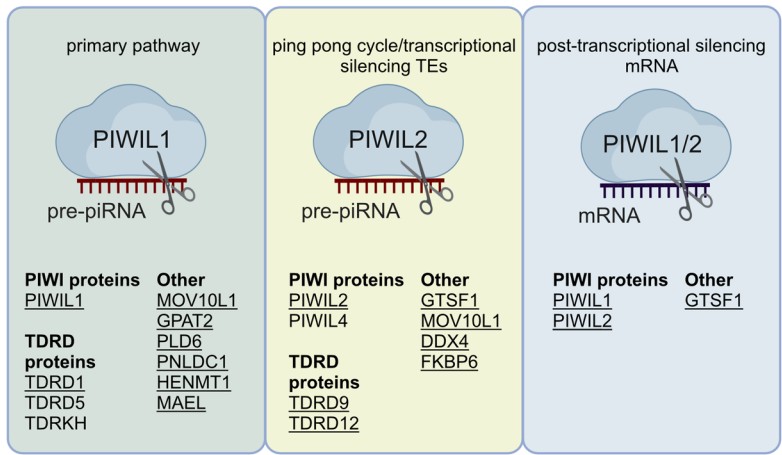

**Fig. 1 | Genetic landscape of piRNA biogenesis-related male infertility.**
**a** Workflow of scrutinizing biological processes related to genetically determined reduced sperm count and male infertility by Gene Ontology (GO) analysis. Pie chart shows first hierarchy of the two-tiered hierarchy. **b** Side-ways bar chart showing -log10(P-value of Bonferroni-adjusted Fisher's Exact test) of individual GO terms. Number of piRNA pathway factors associated with GO term shown to the right of each bar and GO terms associated exclusively with piRNA pathway factors highlighted in blue. **c** Schematic overview on mammalian piRNA biogenesis related sub-pathways with proteins factors known to be involved from mice knockout studies. Proteins in which encoded biallelic high-impact variants were identified in infertile men are underlined. **a**, **c** Created with BioRender.com released under a Creative Commons Attribution-NonCommercial-NoDerivs 4.0 International license (https://creativecommons.org/licenses/by-nc-nd/4.0/deed.en).

**Table 1 | Biallelic high-impact variants identified in genes of the piRNA pathway in infertile men due to azoo-, crypto, or extreme oligozoospermia**

| ID | Gene | Variant (c.) | Variant (p.) | Phenotype (semen; histology TESE outcome) |
|---|---|---|---|---|
| M928 | DDX4 | [1532 C>T];[1532 C>T] | [(Ala511Val)];[(Ala511Val)] | Crypto; RsA |
| M2546[a] | FKBP6 | [508_529dup];[832 C>T] | [(Phe177Cysfs∗20)];[(Arg278∗)] | Crypto; RsA |
| M2548[a] | FKBP6 | [508_529dup];[508_529dup] | [(Phe177Cysfs∗20)];[(Phe177Cysfs∗20)] | Crypto; RsA |
| M1400[a] | FKBP6 | [589-2 A>G];[589-2 A>G] | [(Ala197Glyfs∗31)];[(Ala197Glyfs∗31)] | Crypto; RsA |
| MI-0042P | GPAT2 | [146G>A];[146G>A] | [(Arg49His)];[(Arg49His)] | ExtOligo |
| M2556 | GPAT2 | [1156-1G>A];[1156-1G>A] | [(Glu386Valfs∗16)];[(Glu386Valfs∗16)] | Crypto; MeiA |
| M13 | GPAT2 | [1130A>G];[1954C>T] | [(His377Arg)];[(Arg652∗)] | Azoo; SCO |
| M454 | GPAT2 | [1130A>G];[146G>A] | [(His377Arg)];[(Arg49His)] | Azoo; SCO |
| 17-051 | GPAT2 | [1388C>T];[1388C>T] | [(Thr463Met)];[(Thr463Met)] | Azoo; SCO |
| 15-0730 | GPAT2 | [1388C>T];[1388C>T] | [(Thr463Met)];[(Thr463Met)] | Azoo; SCO |
| M690 | GPAT2 | [1879C>T];[1879C>T] | [(Arg627Trp)];[(Arg627Trp)] | Azoo; MeiA |
| M1844 | GPAT2 | [1879C>T];[1879C>T] | [(Arg627Trp)];[(Arg627Trp)] | Azoo; SCO |
| M2043 | GTSF1 | [97C>A];[97C>A] | [(His33Asn)];[(His33Asn)] | Azoo; MeiA |
| M2243 | GTSF1 | [221_222del];[221_222del] | [(Arg74Lysfs∗4)];[(Arg74Lysfs∗4)] | Azoo; MeiA |
| M3079 | HENMT1 | [400A>T];[400A>T] | [(Ile134Leu)];[(Ile134Leu)] | Azoo; RsA |
| M2435 | MAEL | [799C>T];[908+1G>C] | [(Arg267∗)];[(Cys283_Ala303del)] | Azoo, MeiA |
| TP17 | MOV10L1 | [2179+3A>G];[2179+3A>G] | [(Asn691∗)];[(Asn691∗)] | Azoo; SpgA |
| M1948 | MOV10L1 | [2258T>C];[2258T>C] | [(Val753Ala)];[(Val753Ala)] | Azoo |
| TP24 | MOV10L1 | [3115G>A];[3115G>A] | [(Glu1039Lys)];[(Glu1039Lys)] | Azoo; SCO |
| MI_Proband02199 | MOV10L1 | [3268G>T];[3268G>T] | [(Val1090Phe)];[(Val1090Phe)] | Azoo; SCO |
| M2006 | PIWIL1 | [688C>T];[688C>T] | [(Arg230∗)];[(Arg230∗)] | Azoo; RsA |
| TP32 | PIWIL2 | [839A>C];[839A>C] | [(Tyr280Ser)];[(Tyr280Ser)] | Azoo; SCO |
| M2949 | PIWIL2 | [1697G>A];[1697G>A] | [(Arg566His)];[(Arg566His)] | Azoo |
| M2173 | PLD6 | [1A>T];[1A>T] | [(Met1?)];[(Met1?)] | Azoo; SCO |
| M2803 | PLD6 | [469del];[469del] | [(His157Thrfs∗102)];[(His157Thrfs∗102)] | Azoo; SCO |
| M3274 | PNLDC1 | [790G>T]; [790G>T] | [(Val264Leu)];[(Val264Leu)] | Crypto |
| M1125 | PNLDC1 | [1058A>G];[1058A>G] | [(Tyr353Cys)];[(Tyr353Cys)] | Crypto; ES+ |
| M1648 | TDRD1 | [887C>A];[887C>A] | [(Ser296Tyr)];[(Ser296Tyr)] | Azoo; MeiA |
| M2842 | TDRD9 | [1243G>T];[1243G>T] | [Val415Phe)];[(Val415Phe)] | ExtOligo |
| M800 | TDRD9 | [3148dup];[3148dup] | [(Val1050Glyfs∗49)];[(Val1050Glyfs∗49)] | ExtOligo, ES+ positive TESE |
| M3007 | TDRD9 | [3716+3A>G];[c.3716+3A>G] | [(Ser1208Leufs∗56)];[(Ser1208Leufs∗56)] | ExtOligo |
| M2442 | TDRD9 | [3826 G>T];[3826 G>T] | [(Val1276Phe)];[(Val1276Phe)] | Crypto |
| M2662 | TDRD12 | [287A>C];[287A>C] | [(Asp96Ala)];[(Asp96Ala)] | Azoo; SCO |
| M1642[b] | TDRD12 | [593A>G];[593A>G] | [(Asn198Ser)];[(Asn198Ser)] | Azoo; SCO |
| TP5 | TDRD12 | [963+1G>T];[963+1G>T] | [(Asp289Alafs∗3)];[(Asp289Alafs∗3)] | Azoo; MeiA |
| M2227 | TDRD12 | [986G>A];[986G>A] | [(Trp329∗)];[(Trp329∗)] | Azoo; RsA |
| M2940 | TDRD12 | [2419C>T];[2419C>T] | [(Arg807Cys)];[(Arg807Cys)] | Crypto |
| M2317 | TDRD12 | [2432G>A];[2432G>A] | [(Arg811Gln)];[(Arg811Gln)] | Azoo; ES+ |
| M2595 | TDRD12 | [3157del];[3157del] | [(Leu1053Phefs∗4)];[(Leu1053Phefs∗4)] | Azoo; ES+ |

*Azoo* azoospermia, *Crypto* cryptozoospermia, *ExtOligo* extreme oligozoospermia, *SCO* Sertoli cell-only, *SpgA* spermatogonia arrest, *MeiA* meiotic (spermatocyte) arrest, *RsA* round spermatid arrest, *ES+* elongated spermatids present in seminiferous tubule.
[a]already described[25].
[b]already described[23].

and affects the second histidine of the highly conserved GTSF1 first zinc finger motif (Supplementary Fig. 4c).

**Variants in genes involved in piRNA metabolic processes**
In the primary mammalian piRNA pathway, RNA helicase MOV10L1 selectively binds to cytoplasmic piRNA precursor transcripts[31] and feeds them to the mitochondrial-associated endonuclease PLD6, which catalyzes the first cleavage step of piRNA processing[32].

We identified four azoospermic men with homozygous variants in *MOV10L1* [c.2258T>C p.(Val753Ala) in M1948; c.3115G>A

p.(Glu1039Lys) in TP24; c.3268 G>T p.(Val1090Phe) in TP24; c.2179+3A>G p.? in TP17] (Table 1, Fig. 3a), of which two (TP24, MI_Proband02199) shared a testicular phenotype of SCO, one had sparse spermatogonia (TP17), and the last did not undergo a biopsy (M1948). All three missense variants affect highly conserved amino acids located in α-helical core protein domains, as predicted by AlphaFold2 (Supplementary Fig. 5b). The splice region variant c.2179+3A>G resulted in skipping of *MOV10L1* exon 16 (Supplementary Fig. 5c) and inclusion of a premature stop codon p.(Asn691∗). Two homozygous LoF variants, c.469del p.(His157Thrfs∗102) in M2803 and

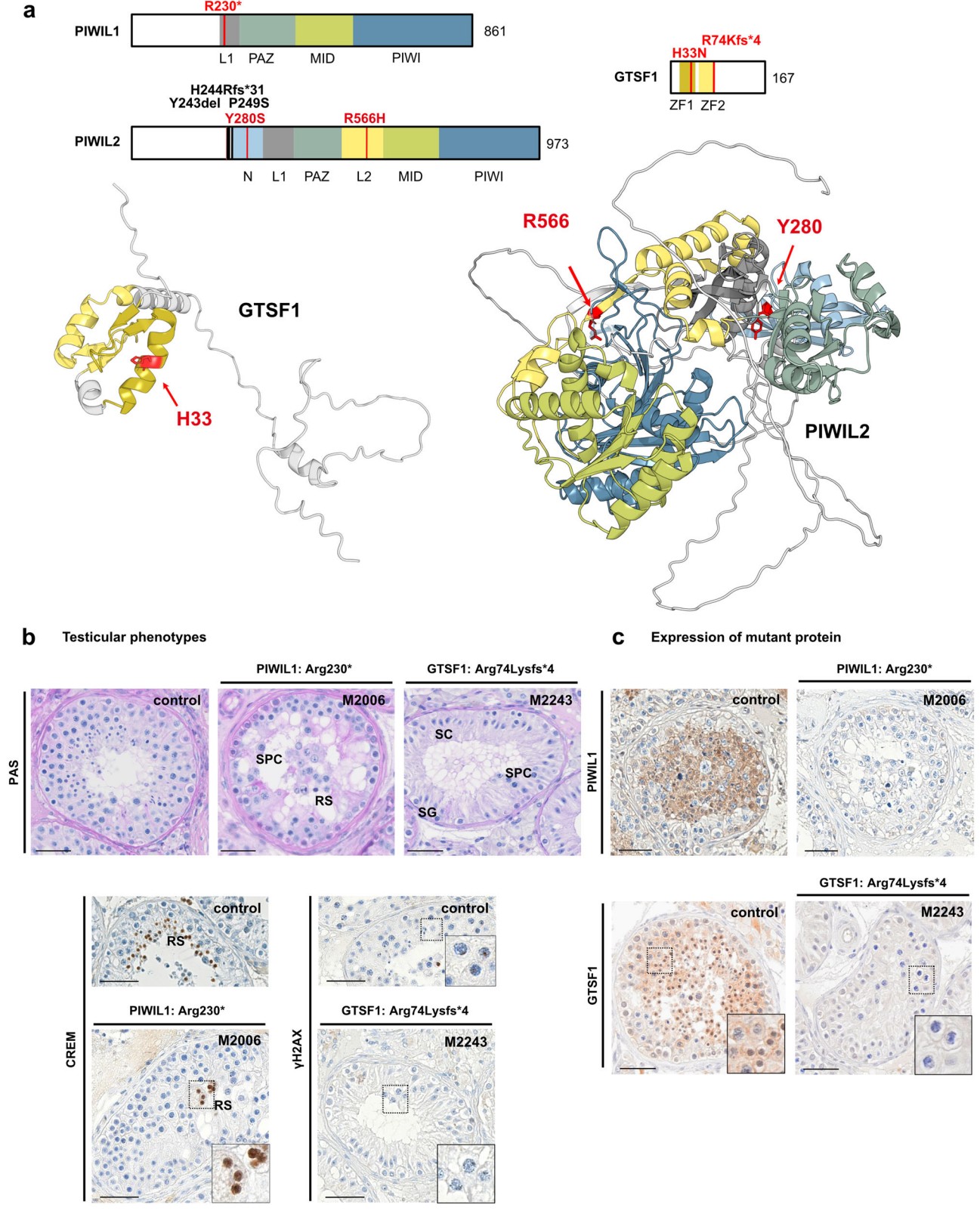

**a**

**b** Testicular phenotypes

**c** Expression of mutant protein

c.1A>T p.(Met1?) in M2173, were identified in *PLD6* in two men with azoospermia due to SCO (Fig. 3, Supplementary Fig. 5d, e). For c.1A>T, the putative loss of translation initiation at the start codon was confirmed by Western blot analysis of HA-tagged PLD6 translated from mutant and wild-type transcripts in HEK293 cells (Supplementary Fig. 5e). This demonstrated that in the mutant, translation starts from downstream in-frame translation initiation sites, resulting in truncated PLD6 proteins lacking at least 129 amino acids at the N-terminus.

PLD6 activity requires a GPAT protein to cleave pre-piRNAs[32] and in mammals, the mitochondrial-associated protein GPAT2 is crucial for primary piRNA biogenesis[33]. Biallelic variants in *GPAT2* were detected in eight men with negative TESE outcomes (Supplementary Table 2; Fig. 3a; Supplementary Figs. 6 and 7a). Five of these variant carriers

**Fig. 2 | Homozygous variants identified in genes of the piRISC complex and associated testicular phenotypes. a** Schematic representation and AlphaFold2 structure predictions of PIWIL1, PIWIL2, and GTSF1. The schematic depicts both novel (red) and known (black) homozygous variants, with amino acids affected by new variants highlighted in the protein structure in red. N N-terminal structured domain, PAZ Piwi/Argonaute/Zwille domain, PIWI Piwi-like domain; L1 linker domain 1, L2 linker domain 2, MID middle domain, ZF zinc finger domain. **b** Periodic acid-Schiff (PAS) staining of testicular tissue of men with normal spermatogenesis (control), M2006 [PIWIL1, p.(Arg230*)] and M2243 [GTSF1, p.(Arg74Lysfs*4)]. Representative tubules showing the most advanced stage of spermatogenesis observed in three independent sections are shown. Immunohistochemical staining (IHC) for round spermatid marker protein Cyclic AMP Element Modulator (CREM) and spermatocyte marker protein γH2AX. In M2006, round spermatids were detected as most advanced germ cells, whereas in M2243, in addition to spermatogonia, rarely pachytene spermatocytes with γH2AX positive sex bodies but no haploid germ cells were observed. **c** IHC staining for PIWIL1 and GTSF1 in controls and variant carriers demonstrating absence of PIWIL1 in M2006 due to homozygous stop-gain variant p.(Arg230*) and absence of GTSF1 in M2243 with homozygous frameshift variant p.(Arg74Lysfs*4). Representative tubules showing the staining pattern observed in independent sections (control: *N* = 3, proband: *N* = 2) are shown. Scale bar = 50 μm. SC Sertoli cell, SG spermatogonia, SPC spermatocyte, RS round spermatid.

share a testicular SCO phenotype, two had meiotic arrest, while only one presented with hypospermatogenesis leading to extreme oligozoospermia (Fig. 3b). The homozygous missense variant c.1879C>T p.(Arg627Trp) was identified in the unrelated patients M690 and M1844, and both parents of M1844 carried this variant in the heterozygous state (Supplementary Fig. 6a). Patients 17-051 and 15-0730 were carriers of c.1388C>T p.(Thr463Met) (Supplementary Fig. 6b). Both originate from Morocco, and a Somalier analysis indicated that they are distantly related. M13 and M454 were compound heterozygous for the missense variant c.1130A>G p.(His377Arg), which is located in the protein's GPAT/DHAPAT acetyltransferase domain and the stop-gain variant c.1954C>T p.(Arg652*) or the missense variant c.146G>A p.(Arg49His), respectively (Supplementary Fig. 6c, d). In M13, p.(His377Arg) was inherited from the mother, who was not carrier of the second variant p.(Arg652*) and p.(Arg49His) was also identified in MI-0042P in the homozygous state. Finally, the splice acceptor variant c.1156-1G>A (M2556) in *GPAT2* resulted in skipping of exon 12 as confirmed by a minigene assay (Supplementary Fig. 7a). This results in a frameshift of the open reading frame and subsequent introduction of a premature stop codon p.(Glu386Valfs*16). Of note, GPAT2 expression was absent in the spermatocytes of the two analyzed *GPAT2* variant carriers (Fig. 3c).

Furthermore, we identified three patients with homozygous missense variants in *PNLDC1* (M3274, M1125) and *HENMT1* (M3079) (Fig. 3a) that both play a crucial role in piRNA maturation[34,35]. The two *PNLDC1* variant carriers exhibited cryptozoospermia and, fittingly, the testicular biopsy of M1125 revealed only a few tubules with elongated spermatids while TESE was negative. (Supplementary Table 2). The missense variants impact two highly conserved amino acid residues, both situated in the PNLDC1 CAF domain (Supplementary Fig. 7b). The homozygous variant c.400A>T p.(Ile134Leu) in *HENMT1* was identified in M3079 affected by azoospermia due to round spermatid arrest (Fig. 3b). The affected tyrosine residue is located in the protein's methyl-transferase domain and is conserved up to zebrafish (Supplementary Fig. 7c).

Finally, we also identified biallelic variants in genes that are limited to secondary biogenesis or post piRNA maturation processes. Among these, *DDX4* encodes a germ cell-specific RNA helicase required for ribonucleoprotein remodeling during the loading of secondary piRNA intermediates onto PIWIL4[19]. In *DDX4*, the homozygous missense variant c.1532 C>T p.(Ala511Val) was identified in an infertile man (M928) with cryptozoospermia due to predominant round spermatid arrest in the testicular tissue (Fig. 3b). Alanine 511 is present in orthologous proteins up to fruit fly and is located in a highly conserved core protein region between the two predicted helicase domains (Supplementary Fig. 8a). However, the cellular expression profile of DDX4 in the patient's testicular tissue remained unchanged (Supplementary Fig. 9a).

The piRNA pathway component MAEL localizes to the cytoplasm and shuttles to the nucleus in round spermatids[36]. It may also facilitate nucleo-cytoplasmic trafficking of PIWIL4–piRNA complexes[37] and pachytene piRNA intermediate processing[38]. M2435 carried the confirmed compound heterozygous stop-gain variant c.799C>T p.(Arg267*) and the splice site variant c.908+1G>C in *MAEL*, which was shown to cause skipping of exon 9 (Supplementary Fig. 8b), resulting in an in-frame deletion of 21 amino acids p.(Cys283_Ala303del). No sperm could be retrieved from the testicular biopsy (Supplementary Table 2) showing pachytene spermatocytes (Fig. 3b) and few CREM-positive haploid germ cells (Supplementary Fig. 3) in single tubules, indicating a spermatogenic arrest at stages downstream of pachytene or after completion of meiosis. Assumed degradation of mutant *MAEL* transcripts by NMD was supported by absence of MAEL-specific staining in spermatocytes and round spermatids in patient testicular tissue sections (Fig. 3d).

## Variants in the scaffold proteins encoded by the TDRD gene family

Tudor domain (TD)-containing proteins (TDRDs) play a crucial role as molecular scaffolds in piRNA biogenesis[39] and, in mice, several members of the TDRD gene family have been linked to piRNA biogenesis. We identified rare homozygous variants in *TDRD1*, *TDRD9*, and *TDRD12* that matched our filtering criteria.

The missense variant c.887C>A p.(Ser296Tyr) in *TDRD1* in M1648 with meiotic arrest affects a highly conserved serine residue located in the first tudor domain (Fig. 4a; Supplementary Fig. 10a). In testicular tissue with complete spermatogenesis, TDRD1 is expressed in perinuclear structures within round spermatids (Supplementary Fig. 9b). Because the seminiferous tubules of M1648 lack haploid germ cells, it remains unknown whether p.(Ser296Tyr) has any effect on the expression or stability of TDRD1.

For *TDRD9*, we identified four infertile men with homozygous nucleotide substitutions predicted to affect the protein sequence: two homozygous LoF variants, c.3148dup p.(Val1050Glyfs*49) in M800 and c.3716+3A>G p.? in M3007, which causes skipping of *TDRD9* exon 32 (Supplementary Fig. 10b, c), resulting in a frameshift p.(Ser1208Leufs*56), and two missense variants, p.(Val415Phe) in M2842 and p.(Val1276Phe) in M2442 (Table 1, Fig. 4a). The affected valine 415 is located in the helicase domain of TDRD9, whereas valine 1276 is located in a C-terminal protein region (Fig. 4a) and both amino acids are conserved in orthologous proteins (Supplementary Fig. 10b). Interestingly, haploid sperm with impaired motility and abnormal morphology were observed in all four *TDRD9* variant carriers (Table 1, Supplementary Table 2).

Finally, in *TDRD12*, seven men with homozygous high-impact variants (Table 1, Fig. 4a) were identified out of whom five had a negative TESE attempt. Two patients with SCO were carriers of the homozygous missense variants c.287A>C p.(Asp96Ala) identified in M2662 and c.593A>G p.(Asn198Ser) identified in M1642, respectively. Both variants co-segregate with the disease in the respective families (Supplementary Fig. 11a,b). TP5 with meiotic arrest carried the homozygous splice site variant c.963+1G>T, leading to skipping of exon 9 (Supplementary Fig. 11c). As a result, the open reading frame is shifted resulting in the synthesis of a truncated protein p.(Asp289Alafs*3), if the mutant transcript is not degraded by NMD.

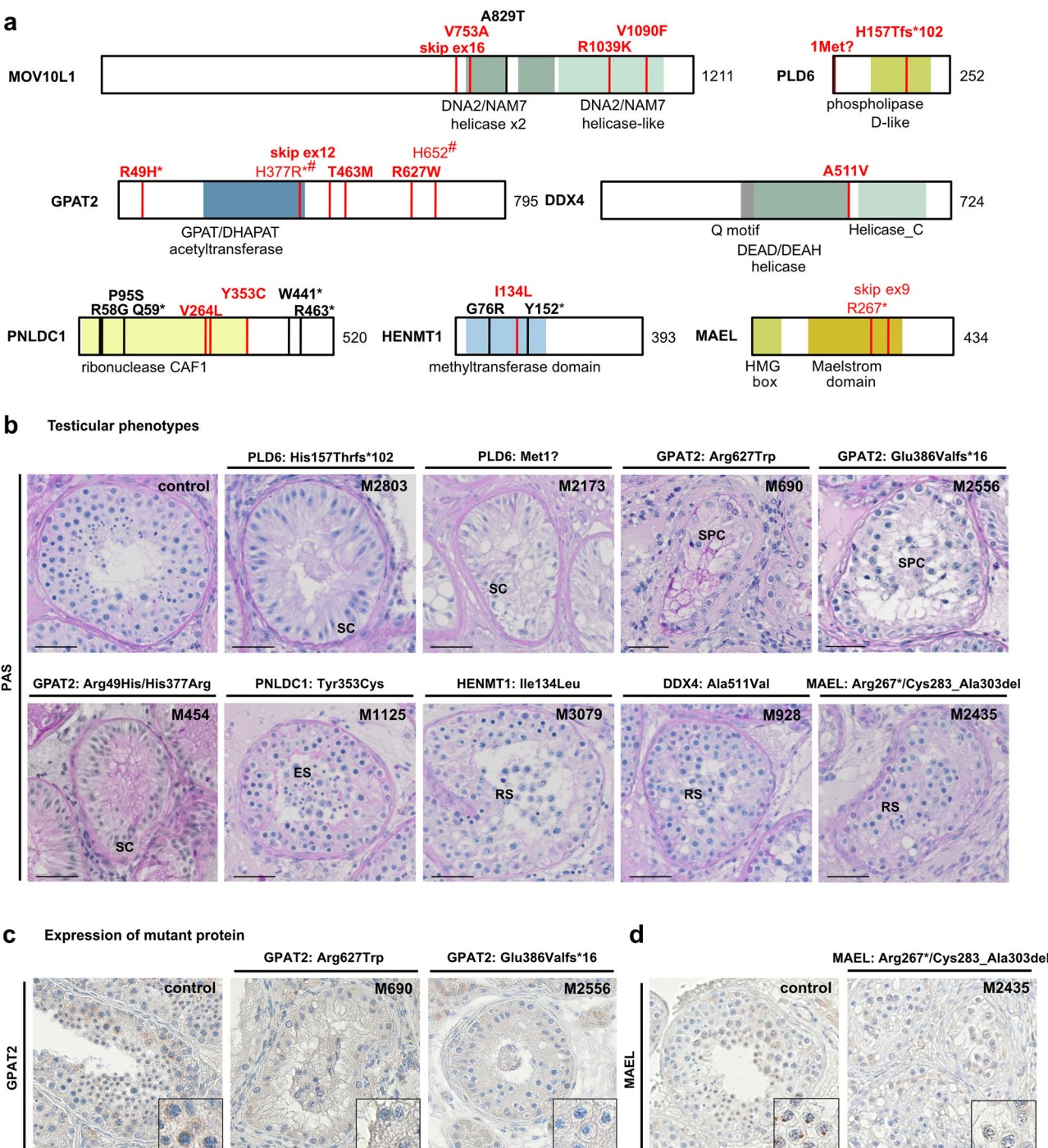

**Fig. 3 | Biallelic variants identified in human piRNA biogenesis-associated genes. a** Localization of variants in schematic of MOV10L1, PLD6, GPAT2, PNLDC1, MAEL, DDX4, and HENMT1 structure with protein domains colored and newly identified biallelic variants (red, bold for homozygous) as well as previously described homozygous variants (black) indicated. Pairs of compound heterozygous variants are indicated by identical symbols (*,#) in superscript. Helicase domains (green): DEAD/DEAH, Helicase_C, DNA2/NAM7; CAF1 chromatin assembly factor 1 domain (yellow); GPAT/DHAPAT acetyltransferase and methyl-transferase domains (blue). **b** Periodic acid-Schiff (PAS) staining of representative testicular tissue of variant carriers demonstrating SCO in M2803 [PLD6, p.(His157Thrfs*102)], M2173 [PLD6, p.(Met1?)] and M454 [GPAT2, p.(His377Arg)/ (Arg49His)] and presence of haploid germ cells (round/elongated spermatids) in M1125 [PNLDC1, p.(Tyr353Cys)], M3079 [HENMT1, p.(Ile134Leu)], M928 [DDX4, p.(Ala511Val)], and M2435 [MAEL, p.(Arg267*)/ p.(Cys283_Ala303del)]. Representative tubules showing the most advanced stage of spermatogenesis observed in three independent sections are shown. **c** Immunohistochemical (IHC) staining for GPAT2 in testicular tissue with full spermatogenesis (control) and *GPAT2* variant carriers with meiotic arrest, [M690, p.(Arg627Trp)], [M2556, p.(Glu386Valfs*16)]. In control tissue, GPAT2 is expressed in perinuclear structures in spermatocytes and this staining pattern is absent in M690 and M2556. Representative tubules showing the staining pattern observed in independent sections (control: $N = 3$, proband: $N = 2$) are shown. **d** IHC for MAEL in testicular tissue with full spermatogenesis (control) and M2435 with compound heterozygous presence of two MAEL LoF variants p.(Arg267*)/p.(Cys283_Ala303del). In control tissue, MAEL is expressed in perinuclear structures in spermatocytes and distinct condensed structures in round spermatids and this staining pattern is absent in the variant carrier. Representative tubules showing the staining pattern observed in independent sections (control: $N = 3$, proband: $N = 2$) are shown. Scale bar = 50 μm. SC Sertoli cell, SPC spermatocyte, RS round spermatid, ES elongated spermatid.

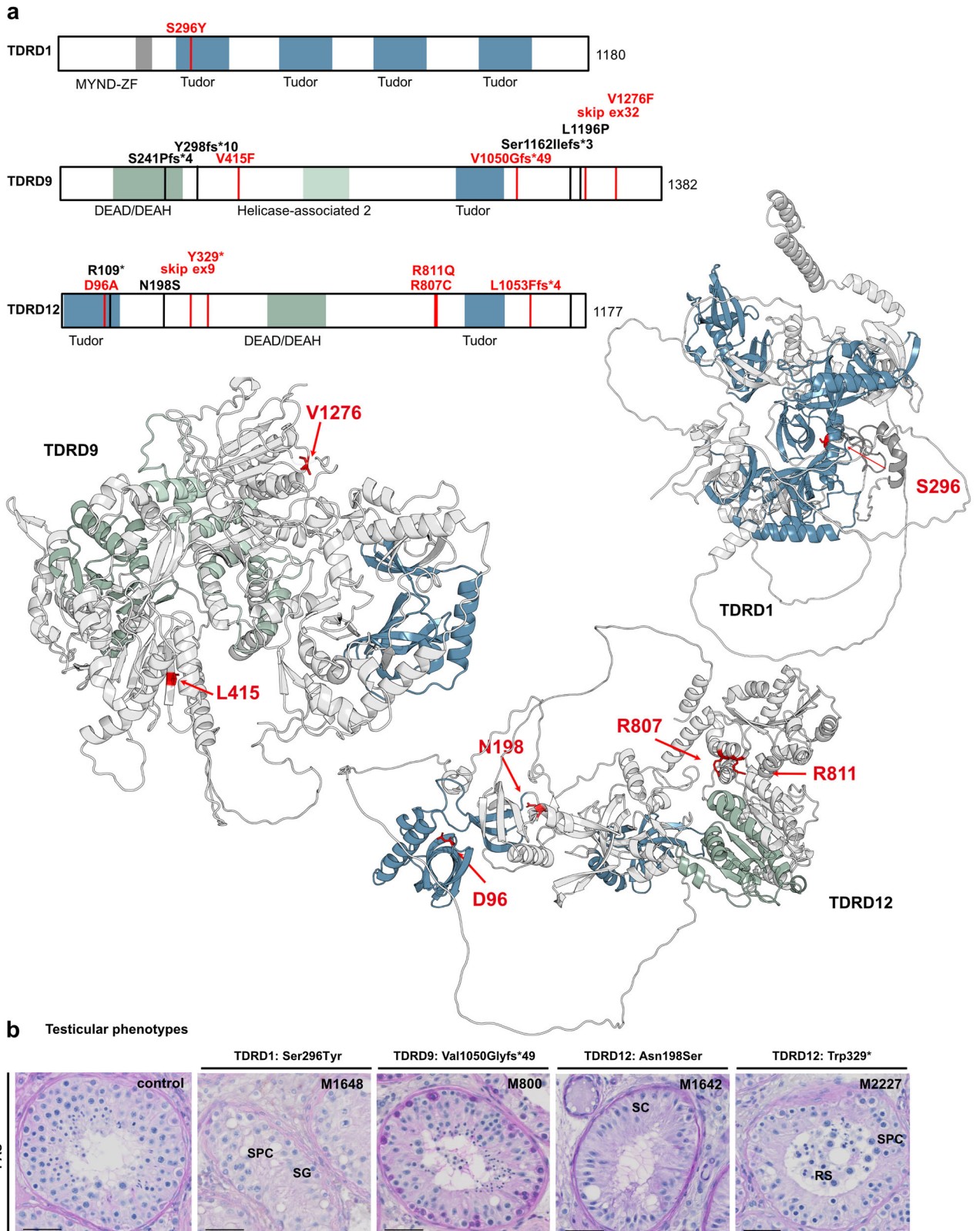

**Fig. 4 | Homozygous high-impact variants identified in human genes of the tudor-domain containing gene family (TDRDs). a** Schematic representation and AlphaFold2 structure predictions of TDRDs. The schematic depicts both novel (red) and known (black) homozygous variants, with amino acids affected by new variants highlighted in the protein structures in red. MYND-ZF MYND-type zinc finger domain (gray), Tudor tudor domain (blue), helicase domains DEAD/DEAH (green), helicase-associated 2 (light green). **b** Periodic acid-Schiff (PAS) staining of testicular tissue of variant carriers demonstrating meiotic arrest in M1648 [TDRD1, p.(Ser296Tyr)], presence of elongated spermatids in M800 [TDRD9, p.(Val1050-Glyfs*49)], SCO in M1642 [TDRD12, p.(Asn198Ser)] and round spermatid arrest in M2227 [TDRD12, p.(Trp329*)]. Representative tubules showing the most advanced stage of spermatogenesis observed in three independent sections are shown. Scale bar = 50 μm. SC Sertoli cell, SG spermatogonia, SPC spermatocyte, RS round spermatid.

Interestingly, in the four other *TDRD12* variant carriers, haploid germ cells (round or elongated spermatids) were detected in the seminiferous tubules (Fig. 4b, Supplementary Fig. 3) or sperm were found in the ejaculate. Two of these patients were carriers of homozygous LoF variants, the stop gain variant c.986G>A p.(Trp329*) in M2227 and the frameshift variant c.3157del p.(Leu1053Phefs*4), in M2595 (Supplementary Fig. 11d, e). Interestingly, a sister of M2227, who is also homozygous for the familial *TDRD12* stop-gain variant, was affected by impaired fertility due to premature ovarian insufficiency. This diagnosis was obtained after she had given birth to a son at the age of 19. The other two *TDRD12* variants, c.2419C>T p.(Arg807Cys) in M2940 and c.2432G>A p.(Arg811Gln) in M2317, result in the substitution of highly conserved arginine residues (Supplementary Fig. 11f) which are found on the surface of a conserved and globular structured region of TDRD12 with unknown molecular function.

### Impact of identified variants on expression of piRNA pathway components

Furthermore, we aimed to explore whether the LoF of one piRNA biogenesis related protein might influence the expression profile of further proteins involved in this metabolic process. To this end, we performed immunohistochemical staining for several key piRNA pathway-related proteins (Fig. 5a, b, Supplementary Figs. 12–15). Indeed, a diminished expression of PIWIL1, PLD6, MAEL, and HENMT1 in spermatocytes was observed in several variant carriers. TDRD1-specific staining was absent only in round spermatids of the *PIWIL1* stop-gain variant carrier M2006, and characteristic and distinct staining of MAEL-positive structures was more diffuse in spermatocytes of the *TDRD1* and *GPAT2* variant carriers. Of note, the staining pattern of DDX4 and GTSF1 was not affected in any of the variant carriers analyzed. Collectively, these observations are an indication for a gene-/protein-specific impact of several of the piRNA biogenesis proteins on the expression of additional piRNA factors.

### Impact of identified variants on piRNA processing and transposon silencing

In the patients with available snap frozen testicular tissue and who were not affected by total germ cell loss [M2006: PIWIL1 p.(Arg230*), M1648: TDRD1 p.(Ser296Tyr), M2595: TDRD12 p.(Leu1053Phefs*4), M2317: TDRD12 p.(Arg811Gln)], we analyzed the impact on piRNA biogenesis in germ cells and performed small-RNA sequencing. The mapped piRNA sequences were intersected with known pachytene piRNA loci detected in the human adult testis. This revealed significantly decreased amounts of piRNAs in all four patients, compared with tissue with complete spermatogenesis (Fig. 5c). Notably, the peak of piRNAs, with a length of 28-31 bases seen in the control tissue was absent in all four samples. Interestingly, the reduction in the amount of pachytene piRNAs observed in the *PIWIL1* stop-gain variant carrier was even more pronounced than the effect seen in previously published piRNA-seq data on *FKBP6* variant carriers even though these patients share a comparable testicular phenotype of round spermatid arrest (Supplementary Fig. 16a).

In mice, disruption of piRNA biogenesis leads to upregulation of transposons. We, therefore, investigated the silencing of transposons in human male germ cells and performed immunohistochemical staining for LINE1 open reading frame 1 protein (LINE1 ORF1p) in the testicular sections of variant carriers. Using a monoclonal, validated antibody directed against human LINE1 ORF1p, no staining was detected in germ cells of human control samples with complete spermatogenesis. In contrast, a concordant and specific expression of LINE1 ORF1p in spermatogonia (Fig. 5d) was observed in three *TDRD12*, two *GPAT2*, three *FKBP6*, and single *MAEL*, *HENMT1*, and *TDRD9* variant carriers, while in all other cases, including carriers of homozygous LoF variants in *PIWIL1* and *GTSF1*, no LINE1 ORF1p was expressed. (Fig. 5d, Supplementary Fig. 16b). This mutant-specific staining pattern was

confirmed by using a second monoclonal LINE1 ORF1p antibody (Supplementary Fig. 16b, c).

In summary, protein expression, pachytene piRNA level, and/or TE expression in 14 variant carriers (10 biallelic LoF, 4 homozygous missense) supported the pathogenicity of the respective variants.

### Comparison of piRNA factor gene-related testicular phenotypes between mice and men

In mammals, detailed information on phenotypic consequences of disturbed piRNA biogenesis are mainly derived from knockout mouse models. We, therefore, compared the testicular phenotypes observed in the affected men with the phenotype of the respective knockout mice. In mice, several of the piRNA factors highlighted in this study have been associated with meiotic arrest when impaired[14,16,18,40–44]. However, in *Piwil1*[45], *Tdrd1*[46], *Pnldc1*[34,47], and *Henmt1*[35] knockout mice, germ cell maturation progressed up to the round or elongated spermatid stage (Fig. 6a), while knockout of *Mael* has been reported to cause meiotic arrest and round spermatid arrest, depending on the genetic background[38,48]. The phenotypic spectrum in humans seems to be broader, ranging from complete absence of sperm as seen in variant carriers of *GPAT2*, *PLD6*, *PIWIL2*, and *TDRD12* to hypospermatogenesis, resulting in severe oligozoospermia, seen concordantly in *TDRD9* variant carriers. In some of the affected men, the testicular phenotype overlaps with the phenotype of the respective knockout mouse model, i.e., round spermatid arrest has been observed in both the human *PIWIL1* LoF variant carriers and the *Piwil1* knockout mice. Interestingly, in *PIWIL2*, *PLD6*, *TDRD1*, and *GPAT2* variant carriers, the phenotype is more severe in men than observed in mice. In contrast, in *TDRD9* and some *TDRD12* variant carriers, the germ cell maturation proceeds to haploid germ cells, i.e., further than in the respective mouse models that exhibit an arrest at meiosis.

## Discussion

The number of identified monogenic causes of male infertility due to impaired spermatogenesis is steadily increasing, and a striking subset of disease genes described encode proteins with vital roles in meiosis[49]. Through comprehensive exploration of biallelic variants in exome/genome data of >2000 infertile men, we provide evidence that, beyond meiosis-related genes, genes encoding proteins involved in piRNA biogenesis are a major, previously underexplored contributor to human spermatogenic failure.

In five of the 14 piRNA genes, namely *PIWIL1*, *GTSF1*, *PLD6*, *GPAT2*, and *MAEL*, biallelic LoF variants were identified in infertile men, introducing them as autosomal recessive disease genes. In this context, the identification of the homozygous LoF variant in *PIWIL1* also resolves the controversy regarding a previously proposed association of heterozygous missense variants in *PIWIL1* with azoospermia, which we had already suspected to be erroneous[50,51]. Furthermore, this study also reports homozygous potentially pathogenic missense variants in *TDRD1* and *DDX4*, both of which are highly intolerant to genetic variations.

Among the genes highlighted in this study, *FKBP6*[25], *PIWIL2*[22,52], *PNLDC1*[23,24], *PLD6*[23], *HENMT1*[20], *MOV10L1*[53], *TDRD9*[20,21,23], and *TDRD12*[23] were recently described in the context of piRNA pathway dysfunction and/or human male infertility and the discovery of additional variants significantly strengthens the gene-disease relationship. For several of these genes, we identified similar testicular phenotypes as previously reported: SCO in *PIWIL2*[52], round spermatid arrest in *PNLDC1*[24], and hypospermatogenesis in *TDRD9* variant carriers[20], respectively. Interestingly, also the gene-specific patients' phenotypes observed in this study were largely consistent and severity was independent from the type of variant, indicating that the identified missense variants are indeed also LoF variants on the protein level (Fig. 6b). Thus, the gene-specific testicular phenotype can be used to aid assessment of the variant's pathogenicity. In contrast, *TDRD12* variant carriers exhibited

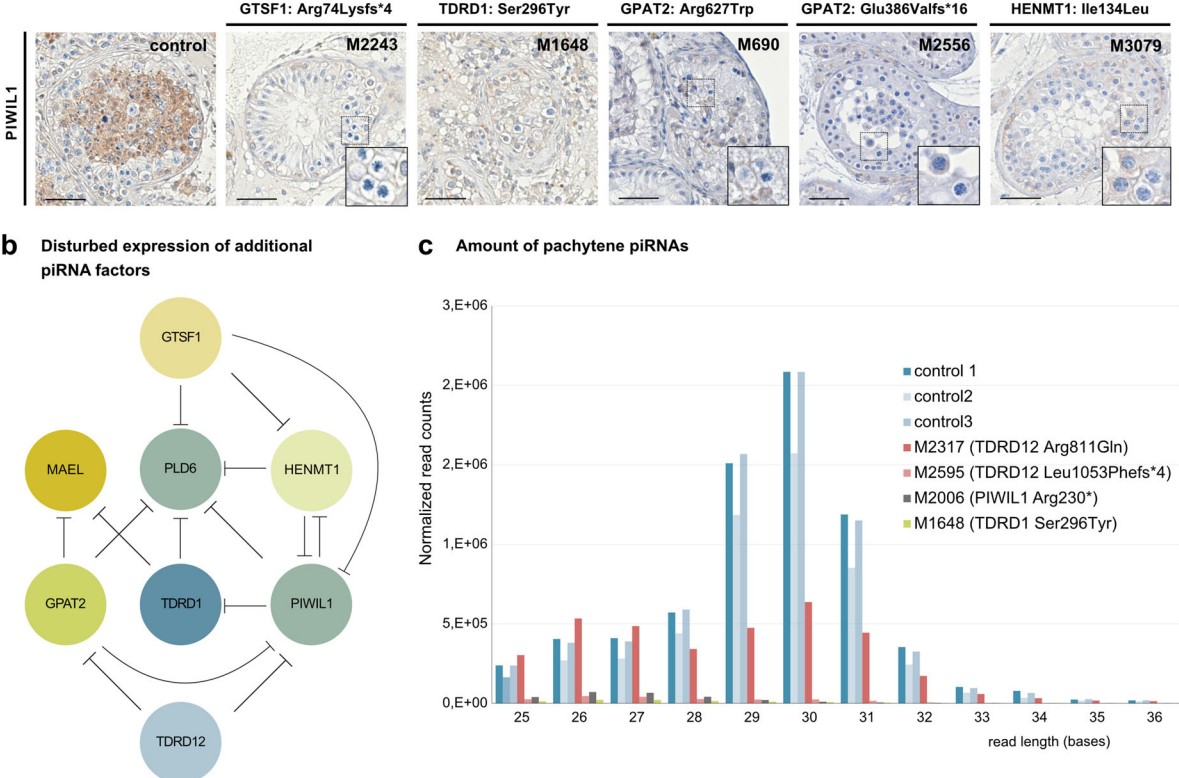

**a** Impact on expression profile of PIWIL1

**b** Disturbed expression of additional piRNA factors

**c** Amount of pachytene piRNAs

**d** Transposon acitivity: LINE1 ORF1p expression

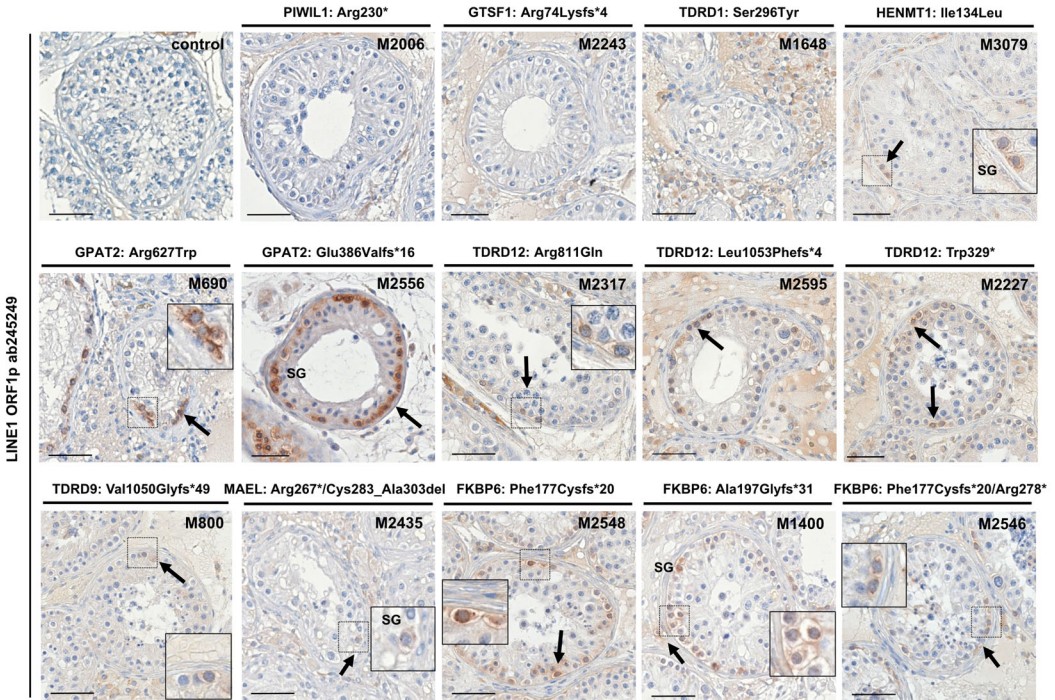

highly variable phenotypes, ranging from SCO to even a few sperm in the ejaculate. In summary, with at least four different biallelic variants (including several LoF variants) identified per gene in this and other studies, *GPAT2*, *PNLDC1*, *TDRD12*, *MOV10L1*, *PLD6*, *FKBP6*, and *TDRD9* are excellent candidates to be included in the diagnostic workup of infertile men.

When comparing the reproductive phenotypes per gene between mice and men, we observed for some genes, including *PIWIL2* and *PLD6*, a depletion of germ cells, while the knockout mice revealed meiotic arrest. Interestingly, aged *Piwil2* knockout mice show a complete lack of germ cells in half of their seminiferous tubuli[54] and in the *Gpat2* knockout mice, apoptosis of germ cells was observed, also

**Fig. 5 | Functional impact of disturbed piRNA biogenesis.**
**a** Immunohistochemical staining demonstrating diminished expression of PIWIL1 in variant carriers M2243 [GTSF1, p.(Arg74Lysfs*4)], M1648 [TDRD1, p.(Ser296Tyr)], M690 [GPAT2, p.(Arg627Trp)], M2556, [GPAT2, p.(Glu386Valfs*16)] and M3079 [HENMT1, p.(Ile134Leu)]. Representative tubules showing the staining pattern observed in independent sections (control: N = 3, proband: N = 2) are shown. **b** Schematic depicting impact of loss of piRNA biogenesis protein function on expression of additional piRNA factors. Circles represent piRNA protein and inhibiting effects of loss-of-protein functions on the expression of further piRNA proteins are indicated. **c** Effect of biallelic variants in genes of the piRNA pathway on biogenesis of pachytene piRNAs. RNA isolated from snap frozen testicular tissue of M2006 [PIWIL1 p.(Arg230*)], M1648 [TDRD1 p.(Ser296Tyr)], M2317 [TDRD12 p.(Arg811Gln)] and M2595 [TDRD12 p.(Leu1053Phefs*4)] revealed a major loss of pachytene piRNAs with expected lengths of 26–31 bases when compared with controls with complete spermatogenesis (ctrl1-3; P = 0.000007). Shapiro-Wilk test was used to estimate the normality of the data. Since Shapiro-Wilk test indicated

abnormal data distribution in both control and case groups, two-sided Mann-Whitney U test was used for comparing the expression changes of piRNAs with different length (26–31 nt) between both groups. Source data are provided as a Source Data file. **d** Immunohistochemical staining for LINE1 transposon specific protein LINE1 ORF1p in testicular tissue. LINE1 ORF1p was not detected in testicular tissue of controls with full spermatogenesis and PIWIL1, GTSF1, and TDRD1 variant carriers. In contrast, all three TDRD12 variant carriers, both GPAT2 variant carriers, and all three FKBP6 variant carriers revealed a concordant distinct and specific LINE1 ORF1p staining in spermatogonia. A similar effect was also seen in testicular tissue of MAEL, HENMT1, and TDRD9 variant carriers. Representative tubules showing the staining pattern observed in independent sections (control: N = 3, proband: N = 2) are shown. Scale bar = 50 μm. SG spermatogonia. **b** Created with BioRender.com released under a Creative Commons Attribution-NonCommercial-NoDerivs 4.0 International license (https://creativecommons.org/licenses/by-nc-nd/4.0/deed.en).

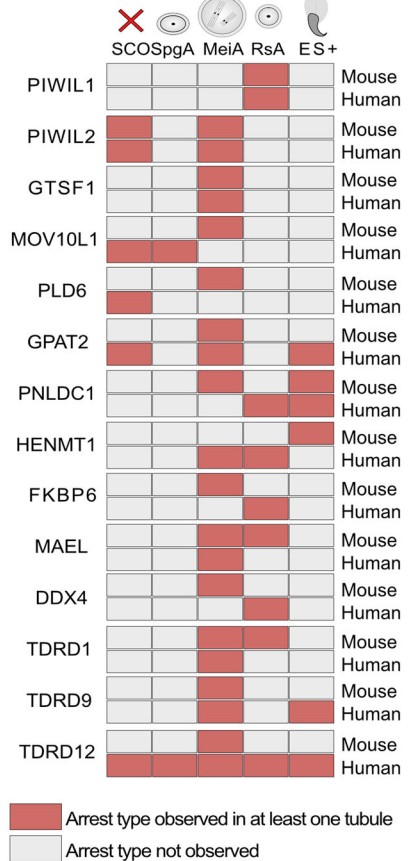

**a** Testicular phenotypes in human and mice

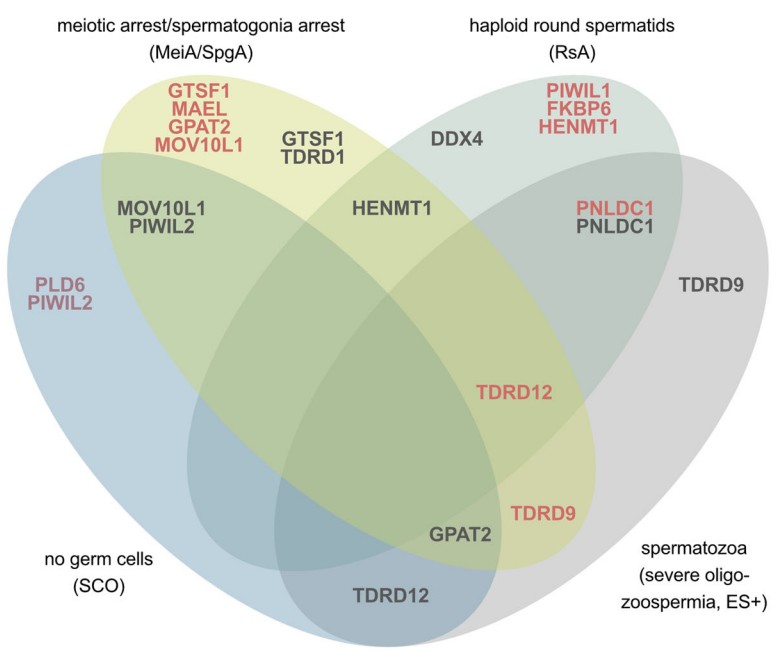

**b** Reproductive phenotypes of variant carriers

**Fig. 6 | Comparison of piRNA factor-related male reproductive phenotypes.**
**a** For each piRNA factor gene described in this study, the reproductive phenotype of the male knockout mice is compared with the phenotypes observed in novel and known human infertile biallelic variant carriers. Of note, for several genes, testicular phenotypes of human variant carriers differ from those described for the respective knockout mice. **b** Venn diagram depicting overlap between reproductive phenotypes of infertile men affected by biallelic variants in the same gene. LoF

variants are indicated in red, missense variants are indicated in black. For several genes, (PIWIL2, GTSF1, MOV10L1, PNLDC1, TDRD9) a phenotypic overlap could be observed for missense and LoF variant carriers. LoF variant carriers are in most cases not affected by a more severe phenotype than missense variant carriers. SCO Sertoli cell-only, SpgA spermatogonia arrest, MeiA meiotic arrest, RsA round spermatid arrest, ES+ elongated spermatids present in testicular tissue.

supporting an age-dependent progressive loss of germ cells[16]. However, the more severe phenotype of SCO observed in the human variant carriers of PLD6, GPAT2, PIWIL2, and TDRD12 was not associated with increased patient age. In contrast, carriers of FKBP6 or TDRD9 presented a less severe phenotype of round spermatid arrest or even hypospermatogenesis resulting in severe oligozoospermia, while the

corresponding knockout mice showed meiotic arrest[43,55]. Since for many of these cases, it has not been fully proven that the variants lead to a complete loss of protein function, further data is needed to draw firm conclusions whether, in the case of some piRNA factor genes, human spermatogenesis may be less stringently controlled and progresses further despite disrupted piRNA biogenesis. It remains to be

determined whether the round spermatids/sperm produced in some men are actually suitable for procreation.

For several of the identified variants, we demonstrate functional data linking impaired function or absence of the encoded protein to downstream cellular effects. The observed diminished expression of further piRNA factors in variant carriers was also described for *PNLDC1*[24], where it has been linked to decreased expression of *MYBL1*, a testis-specific transcription factor known to regulate expression of pachytene piRNAs as well as several piRNA factor genes in mice and men[5,56]. However, due to the limited amount of testicular tissue from variant carriers available for analysis of piRNA factor expression, the data presented here are a first indication of a co-dependency and it cannot be ruled out that different expression profiles are at least in part also related to different germ cell compositions of the testicular sections.

In several piRNA factor knockout mice, including *Piwil1* and *Piwil2*, the impaired piRNA biogenesis resulted in de-repression of TEs in spermatocytes of the adult testis[10]. Surprisingly, we observed a spermatogonia-specific de-repression of LINE transposons in *GPAT2*, *TDRD9*, *TDRD12*, *FKBP6*, *HENMT1*, and *MAEL* variant carriers, while on the contrary, homozygous LoF variant carriers in *PIWIL1* and *GTSF1* lacking the encoded protein, did not demonstrate TE de-repression. A spermatogonia-specific de-repression of LINE1 was recently also reported for an azoospermic patient carrying a homozygous LoF variant in *SPOCD1*[57], encoding a crucial protein for piRNA-directed de novo DNA methylation in prospermatogonia[58]. In mice and men, pachytene piRNAs represent more than 90% of all piRNAs in the adult testis and for both species, it has been demonstrated that this subtype of piRNAs binds to PIWIL1 (MIWI), that is specifically expressed from the pachytene stage of meiosis up to the elongated spermatid stage[5,59]. However, in contrast to the highly conserved piRNA biogenesis genes, the pachytene piRNA loci themselves are highly divergent between mice and men.

In humans, it has been shown that the exons of pachytene piRNA precursors are depleted of transposons[5] and this could explain why loss of PIWIL1 in humans does not correlate with enhanced LINE1 ORF1p expression, although the amount of mature pachytene piRNAs is reduced. Accordingly, our data underline that de-repression of transposons does not seem to be a general consequence of impaired pachytene piRNA biogenesis. This conclusion is supported by recent findings in a *Piwil1* N-terminal Arginine-Glycine (RG) motif mutant mouse model demonstrating impaired binding to TDRD proteins, spermatogenic arrest, and reduced levels of pachytene piRNAs[60,61] while LINE1 transposons are still effectively silenced[60]. Because impaired biogenesis of pachytene piRNAs also affects spermatogenic gene expression[6,62], the dysfunctions in spermatogenesis might not result from harmful transposon expression but could be a consequence of transcriptional dysregulation. In other human piRNA genes such as *GPAT2*, *FKBP6*, and *TDRD12*, genetic variants concordantly result in LINE1 de-repression in spermatogonia. Here, the encoded proteins might also be involved in biogenesis of pre-pachytene piRNAs that are mainly loaded to PIWIL2, which is expressed at all stages of male germ cell maturation including spermatogonia. Accordingly, impaired biogenesis of pre-pachytene piRNAs might lead to de-silencing of transposons in spermatogonia resulting in expression of LINE1 ORF1p.

By demonstrating several differences in the consequences of piRNA biogenesis dysfunction between humans and mice, this study highlights that although the piRNA pathway is highly conserved, not all data obtained in mouse models can be readily extrapolated to humans. Recent studies on piRNA pathway dysfunction in golden hamsters already revealed phenotypic discrepancies between different mammalian species[61]. In the golden hamster, pachytene piRNA generation starts earlier than in mice, and dysfunction of PIWIL1, PIWIL2, PIWIL4, and MOV10L1 caused more severe defects in spermatogenesis.

Furthermore, loss of PIWIL1 and MOV10L1 not only led to male but also female infertility[61,63]. It was speculated that among other reasons, these differences might be related to the presence of a fourth PIWI protein, encoded by *Piwil3* in the golden hamster[63]. Interestingly, also humans encode a PIWIL3 protein and we identified one female variant carrier with a homozygous LoF variant in *TDRD12*, diagnosed with infertility due to premature ovarian insufficiency, a phenotype related to impaired oocyte maturation. While in the golden hamster it was shown that PIWIL3 is important only for female fertility[61], the function of this protein in human piRNA biogenesis and fertility still needs to be elucidated.

In conclusion, this study provides extensive data linking disrupted piRNA biogenesis to human spermatogenic failure, demonstrates that piRNA pathway genes are a major target for scrutinizing genetic causes of male infertility, and suggests that safeguarding of the genome during spermatogenesis is in some instances less stringent in men than in mice. The detailed characterization of pathogenic human variants provides insight into the molecular function of the factors involved in piRNA biogenesis and piRNA-mediated transposon silencing. This opens the possibility to investigate key protein domains and, in parallel, to assess the pathogenicity of gene variants.

## Methods

### Ethical approval

All persons gave written consent compliant with local requirements. The study protocol was approved by the local ethics committees: MERGE Münster (2010-578-f-S) and Gießen (26/11); Strasbourg (CPP 09/40−WAC-2008-438 1W DC-2009-I 002), and Yeni Yüzyıl University, Scientific, social and noninterventional health sciences research ethics committee, Istanbul, Turkey (approval no: 2019/08); Barcelona: (2014/04c); Newcastle: (Newcastle:REC ref. 18/NE/0089), Nijmegen: (NL50495.091.14 version 5.0). All experiments were performed in accordance to the criteria set by the Declaration of Helsinki[64].

### Study cohorts

Four cohorts of exome or genome sequencing data of infertile men were included in this study. The MERGE cohort includes data of 2412 men (average age: 34; 2352 exomes and 60 genomes) with various infertility phenotypes and >90% of these men were recruited at the Centre of Reproductive Medicine and Andrology (CeRA), Münster. Most men of this cohort are azoospermic, (HPO:0000027; $N = 1448$) or have severely reduced sperm counts: $N = 454$ with cryptozoospermia (HPO:0030974; sperm only identified after centrifugation of the ejaculate); $N = 158$ with extreme oligozoospermia (HPO:0034815; sperm count < 2 million); $N = 67$ with severe oligozoospermic (HPO:0034818; sperm count <10 million). Numerical chromosomal aberrations such as Klinefelter syndrome (karyotype 47, XXY) and Y-chromosomal AZF-deletions are exclusion criteria. Likely pathogenic monogenic causes for the infertile phenotype were already described in about 8% of cases[26].

The Strasbourg cohort comprises 23 men diagnosed with NOA. The Barcelona cohort (BCN) comprises 235 NOA men attending the Fundació Puigvert (Barcelona)[65]. The Nijmegen/Newcastle cohort includes 266 infertile men, 225 affected by azoospermia, and 41 by cryptozoospermia[66].

### Exome and genome sequencing

Sequencing and bioinformatics analyses in the MERGE cohort were performed as previously described[26]. In brief, genomic DNA was extracted from peripheral blood leukocytes via standard methods. For exome sequencing of the MERGE and Strasbourg cohort, the samples were prepared and enrichment was carried out according to the protocol of either Agilent's SureSelectQXT Target Enrichment for Illumina Multiplexed Sequencing Featuring Transposase-Based Library Prep

Technology (Agilent) or Twist Bioscience's Twist Human Core Exome. To capture libraries, Agilent's SureSelect Human All Exon Kits V4, V5, and V6 or Twist Bioscience's Human Core Exome plus RefSeq spike-in and Exome 2.0 plus comprehensive spike-in were used. For whole genome sequencing of samples from the MERGE cohort sequencing libraries were prepared according to Illumina's DNA PCR-Free library kit. For multiplexed sequencing, the libraries were index tagged using appropriate pairs of index primers. Quantity and quality of the libraries were determined with the ThermoFisher Qubit, the Agilent TapeStation 2200, and Tecan Infinite 200 Pro Microplate reader, respectively. Sequencing was performed on the Illumina HiSeq 4000 System, the Illumina HiSeqX System, the Illumina NextSeq 500 System, the Illumina NextSeq 550 System, or the NovaSeq 6000 System, using the HiSeq 3000/4000 SBS Kit (300 cycles), the HiSeq X Ten Reagent Kit (300 cycles), the NextSeq 500 V2 High-Output Kit (300 cycles), or the NovaSeq 6000 S1 and S2 Reagent kits v1.5 (200 cycles), respectively. For the BCN cohort exome sequencing was carried out as a service by Macrogen Inc. 122 (Republic of Korea) using the 123 Agilent SureSelect_V6 enrichment and a NovaSeq 6000. Exome sequencing in the Nijmegen/Newcastle cohort was performed as previously described[66]. Briefly, samples were prepared and enriched following the manufacturer's protocols of either Illumina's Nextera DNA Exome Capture kit or Twist Bioscience's Twist Human Core Exome Kit and sequencing was performed on the Illumina NovaSeq 6000 Sequencing System.

### Variant calling
After trimming of remaining adapter sequences and primers with Cutadapt v1.15[67], reads were aligned against GRCh37.p13 using BWA Mem v0.7.17[68]. Base quality recalibration and variant calling were performed using the GATK toolkit v3.8[69] with haplotype caller according to the best practice recommendations. For more recent samples and whole genome samples Illumina Dragen Bio-IT platform v4.2 was used for alignment and variant calling. Both pipelines use GRCh37.7.p13 as reference genome. Resulting variants were annotated with Ensembl Variant Effect Predictor[70]

### Gene Ontology analysis
Exome data of infertile men from MERGE was first filtered for genes with rare (MAF≤0.01 according to the gnomAD, v2.1.1) homozygous LoF variants (stop-gain, start-loss, splice acceptor, splice donor, frameshift). We then selected for genes preferentially expressed in human male germ cells according to single cell RNAseq data included in the human protein atlas (HPA)[71]. GO analysis (http://geneontology.org)[72–74] was performed on this gene list (for "biological processes" and "homo sapiens") and processed with PANTHER https://pantherdb.org/webservices/go/overrep.jsp[74,75] (annotation dataset: "GO biological processes complete", test type: 'Fisher's Exact', Correction: "Bonferroni", showing results with $P < 0.05$). GO terms were then processed with Revigo[76] (http://revigo.irb.hr/) using the $P$-value and a medium (0.7) list setting (yes to removal of obsolete GO terms, species "homo sapiens", "SimRel" semantic similarity measure). The Revigo Table was exported and -log10($P$ value) of representative GO terms (classed as representation: "null") plotted as side-ways bar chart. Revigo tree data was processed with CirGO.py[77] for visualization of the 2-tiered hierarchy of GO-terms.

### Screening of exome data for biallelic high-impact variants
To identify potentially harmful gene variants in genes of the piRNA pathway, exome data of infertile men from all cohorts included in this study were screened to identify individuals with biallelic high-impact variants (stop-gain, start-loss, splice site and splice region, deletions, and insertions as well as missense variants with CADD ≥ 15) in a total of 24 different genes of the pathway (Table S1). Only variants with a MAF ≤ 0.01 (gnomAD database, v 2.1.1) were taken into account.

To exclude the presence of additional possibly pathogenic variants, exome data were additionally screened for additional rare homozygous high-impact variants (LoF and missense variants with CADD ≥ 20) occurring in a list of 21 azoospermia-associated genes with at least moderate clinical validity[26] and 363 candidate genes associated with the GO classification "male infertility" in the Mouse Genome Informatics Database revealing strong expression in human male germ cells. Patients in which additional candidate variants were identified were excluded from further analysis.

### Further genetic analysis
Validation of prioritized variants as well as co-segregation analyses were performed by Sanger sequencing. The regions of interest were amplified from patients genomic DNA as well as available family members with primers and conditions as listed in Supplementary Table 3. The PCR products were then purified and sequenced using standard protocols. For validation of variants in *GPAT2*, long range PCR products using *GPAT2* specific primers, which do not bind to the pseudogenes *GPAT2P1* and *GPAT2P2*, were amplified and used as template for subsequent nested PCR and Sanger sequencing. If a variant was found in more than one individual in MERGE (*GPAT2*: c.1879C>T in M690 and M1844 and c.1130A>G in M13 and M454), the relationship between the two carriers was determined using the Somalier tool[78]. In case that no parental DNA was available for analysis, biallelic occurrence of heterozygous variants was determined by long-read sequencing using long-range PCR products encompassing both genomic regions of interest amplified from variant carriers as template for library generation.

### NGS library preparation and long read sequencing using the MinION system
To determine if two heterozygous variants identified in one gene of the same patient occur in *cis* or in *trans*, a long read sequencing approach was used. To this end, a long-range PCR product encompassing both variants was amplified (see Supplementary Table 3 for primer information) from patients' genomic DNA using the TAKARA LA Taq® DNA Polymerase Hot-Start Version. 1 µg of PCR products was used for subsequent preparation of MinION sequencing library. Barcoding and sequencing was carried out according to manufacturer's instructions (MinION, Oxford Nanopore Technologies). After demultiplexing of obtained reads, alignment to human reference hg19, quality control and variant calling phasing of variants on same/different alleles was determined.

### Minigene assay
To determine the functional impact of splice site and splice region variants, an in vitro splicing assay based on a minigene construct was performed. The region of interest was amplified from genomic DNA of the respective patient as well as of a human control sample by standard PCR procedures. Primers are indicated in Supplementary Table 3. To analyze splice effect of variants *GPAT2* c.1156-1G>A, *MAEL* c.908+1G>C, and *TDRD9* c.3716+3A>G, products were cloned into pENTR™/D-TOPO® according to manufacturer's instructions. The subsequent gateway cloning was performed using Gateway™ LR Clonase™ Enzyme Mix and pDESTsplice as destination vector (pDESTsplice was a gift from Stefan Stamm (Addgene plasmid #32484)[79]. To analyze the *TDRD12* c.963+1G>T variant, the region encompassing exon 8–10 of *TDRD12* was amplified and subcloned into pcDNA3.1 and for *MOV10L1* c.2179+3A>G into pSPL3B. A transient transfection with mutant and wild-type Minigene constructs was performed using Human Embryonic Kidney cells (HEK293T Lenti-X, Clontech Laboratories, Inc.®; catalog number: 632180). Total RNA was extracted using the RNeasy Plus Mini Kit (QIAGEN®) and reverse-transcribed into cDNA with the ProtoScript® II First Strand cDNA Synthesis Kit (New England Biolabs GmbH®). Amplification of the region of interest was performed and

RT-PCR products were separated on a 2% agarose gel, cut out, extracted, and sequenced.

## Characterization of translation initiation in *PLD6* c.1A>T

For cloning of the *PLD6* expression construct pcDNA3.1-*PLD6*-HA, total RNA from human adult testis (BioCat, Heidelberg, Germany) was converted to cDNA using the GoScript™ Reverse Transcriptase system (Promega, Madison, USA). *PLD6* open reading frame (NM_178836.4) with adjacent 3′ and 5′ untranlated regions was amplified from cDNA using PrimeSTAR Max polymerase (Takara Bio, Kusatsu, Japan) and subcloned into the expression vector pcDNA3.1(+) (Genscript, Leiden, NL) followed by insertion of C-terminal HA tag to the *PLD6* open reading frame. Variant c.1A>T was introduced by site-directed mutagenesis using the QuickChange II XL mutagenesis kit (Agilent, catalog number #200522). Primer information are indicated in Supplementary Table 3. Wild-type and mutant constructs were verified by Sanger sequencing.

HEK293 cells were transfected with WT and mutant pcDNA3.1-PLD6-HA using the K2® Transfection Reagent (Biontex). 48 h after transfection cells were washed with ice-cold PBS, scrapped of the plate in 0.8 ml lysis buffer (25 mM HEPES, 100 mM NaCl, 1 mM CaCl2, 1 mM MgCl2, 1% TritonX-100, 1x protease inhibitor cocktail) and lysed for 30 min at 4 °C. 10 µl of cleared lysates were separated on a 4–15% TGX Stain-Free polyacrylamide gel (Mini-PROTEAN, Bio-rad) and transferred to PVDF membrane using Trans Blot Turbo System (Bio-rad). After blocking in 5% milk–TBST, membranes were incubated overnight at 4 °C with anti-HA-tag and anti-GAPDH antibody (Supplementary Table 4). Membranes were washed with TBST and incubated for 1 h with respective HRP-bound secondary antibodies. After washing with TBST membranes were imaged using the ChemiDoc MP Imaging system (Bio-Rad).

## AlphaFold2 protein structure

AlphaFold2 structure predictions were obtained from EBI, except for PNLDC1 and GPAT2, which were generated with the AlphaFold2 Google colab (https://colab.research.google.com/github/sokrypton/ColabFold/blob/main/AlphaFold2.ipynb)[80,81] using protein sequences encoded by the NCBI accession NM_001271862.2 for PNLDC1 and NM_001321526.1 for GPAT2. pdb files of these protein structures are provided as Supplementary Data 2 and Supplementary Data 3. Images of protein structures were generated with Pymol (v.2.5.4, Schrödinger, LLC).

## Histology and Immunohistochemical staining

Testis biopsies of patients from the MERGE cohort and control subjects were obtained from testicular sperm extraction (TESE) approaches at the Department of Clinical and Surgical Andrology (University Hospital Münster, Germany). Biopsies were fixed in Bouin's solution overnight, washed with 70% ethanol and embedded in paraffin for routine histological evaluation. Subsequently, 5 µm sections were stained with periodic acid-Schiff (PAS) according to previously published protocols[82]. In brief, sections were dewaxed in solvent (ProTaqs Clear, #4003011; Quartett Immunodiagnostika and Biotechnologie, Berlin, Germany), rehydrated in a decreasing ethanol series and then incubated for 15 min in 1% periodic acid. After washig with dH2O sections were incubated for 45 min with Schiffs reagent (Roth, Karlsruhe, Germany). Testis biopsies of patients from Gießen were processed equally but stained with hematoxylin and eosin (HE) following previously published protocols[83]. Briefly, dewaxed and rehydrated 5 µm sections were stained for 3 min in hematoxylin, washed for 15 min in dH2O followed by 10 dips in 95% EtOH, and stained for 30 sec with eosin. Testis biopsies of patients from Barcelona were treated as described previously[65]. In brief, 6 µm sections were deparaffinized and rehydrated as follows: 2 × 10 min xylene, 2 × 5 min absolute EtOH,

1 × 2 min 90% EtOH, 1 × 70% EtOH and then washed in dH2O. Tissue sections were stained with hematoxylin for 8 min and washed in running tap water for 10 min. Slides were subsequently rinsed in dH2O followed by 10 dips in 95% EtOH. Following the wash step, tissue sections were counterstained with eosin for 1 min and dehydrated. Finally, the slides were cleared for 2 × 5 min in xylene and mounted with Pertex® mounting medium (Histolab #00801).

For immunohistochemical analyses, 3 µm sections of testicular tissue were de-paraffinized and rehydrated as described[84]. Briefly, paraffin sections were dewaxed in solvent (ProTaqs Clear, #4003011; Quartett Immunodiagnostika and Biotechnologie, Berlin, Germany), rehydrated in a decreasing ethanol series. After rinsing with tap water (15 min, heat-induced antigen retrieval was performed in HIER buffer (pH 6) or as indicated in Supplementary Table 4. This step was followed by cooling and washing with 1X Tris-buffered saline (TBS) before endogenous peroxidase activity was blocked using 3% hydrogen peroxide (15 min, RT). Blocking was performed by adding 25% goat serum (#ab7481, Abcam, UK) in TBS containing 0.5% bovine serum albumin (BSA, #A9647, Merck, Germany, 30 min, RT). Sections were incubated overnight at 4 °C in primary antibody solution, including 5% BSA/TBS and primary antibody as indicated in Supplementary Table 4. The following day, sections were washed with 1x TBS and incubated with a corresponding biotinylated secondary antibody in 5% BSA/TBS for 1 h. After washing with TBS, sections were incubated with streptavidin-horseradish peroxidase (#189733, Merck, Germany– 1:500, 45 min, RT) diluted in 5% BSA/TBS. Subsequently, sections were washed with TBS and incubated with 3,3′-Diaminobenzidine tetrahydrochloride (DAB, #D5637, Merck, Germany) for visualization of antibody binding. Staining was validated by microscopical acquisition and stopped with aqua bidest. Counterstaining was conducted using Mayer's hematoxylin (#109249, Merck, Germany). Finally, sections were rinsed with tap water, dehydrated with increasing ethanol concentrations and mounted using M-GLAS® mounting medium (#103973, Merck, Germany). In each experiment, sections from testicular tissue with full spermatogenesis were included as positive controls as well as omission and IgG controls. In case the proband testicular staining pattern for a respective antibody differed from the staining pattern in the positive control, the the experiment was repeated at least once.

Slides were evaluated and documented using a PreciPoint O8 Scanning Microsocope, Olympus BX61VS Virtual Slide System Axioskop (Zeiss, Oberkochen, Germany), or an Olympus BX61 microscope with an attached Retiga 400R camera (Olympus, Melville, NY, USA) and integrated CellSens imaging software (Olympus, Melville, NY, USA).

## RNA extraction and small RNA sequencing

RNA from snap-frozen testicular tissues of three controls with full spermatogenesis and infertile men with biallelic variants in *PIWIL1* (M2006), *TDRD1* (M1648), *TDRD12* (M2317, M2595) and *FKBP6* (M2546, M2548) was extracted using Direct-zol RNA Microprep kit (Zymo Research, #R2062). The quantity and quality of the isolated RNA were assessed with Qubit RNA High Sensitivity kit (Invitrogen, Cat. #Q32852) and Agilent RNA Nano kit (Agilent, Cat. #55067-1512), respectively.

300 ng of total RNA was used for small RNA library preparation using NEXTflex Small RNA-Seq Kit v3 (PerkinElmer, #NOVA-5132-05). In addition to the manufacturer's protocol, a spike-in mix of 0.05 ng 5′P-cel-miR-39-3p-3′-OH and 0.05 ng 5′P-ath-159a-3′-2-OMe was added at the initial library preparation step, to check for technical errors at library preparation and sequencing steps. Sequencing was carried out at the Oregon Health & Science University Massively Parallel Sequencing Shared Resource facilities on Illumina NovaSeq 6000 S4 2 × 100

flow cell. For RNA-seq data processing and piRNA annotation sequencing reads were trimmed with Cutadapt (v.#3.0) according to instructions provided by CATS small RNAseq kit protocol (Diagenode, #C05010040, Doc. # v.2 I 09.17) or NEXTflex Small RNA-Seq Kit protocol (PerkinElmer, #NOVA-5132-05, Doc. # v.V19.01). Next, trimmed reads were aligned to reference genome (GRCh37) with Bowtie (v.#1.0.1)[85] allowing only perfect matches, discarded miRNAs by selecting reads between 25 and 45 bases, and re-aligned to GRCh37 allowing one mismatch. Finally, known small non-coding RNAs, other than piRNAs, were removed from the dataset using DASHRv2 (v.#v2)[86] and the remaining piRNA sequences were intersected with known piRNA loci detected in human adult testis[3]. For statistical analysis data from small RNA-seq experiments were evaluated using SciPy (ver.: 1.8.0) packages[87]. Shapiro-Wilk test for normality of the data and Mann-Whitney U test was used for comparing the expression changes in piRNA quantities of different lengths (26-31 nt).

## Statistics and reproducibility
Statistical comparisons between two groups were performed by Student's *t* test or Mann-Whitney U test. Experimental replicates were performed as indicated in the respective Figure legends. All putative pathogenic variants were validated by Sanger Sequencing. The Investigators were not blinded to allocation during experiments and outcome assessment.

## Reporting summary
Further information on research design is available in the Nature Portfolio Reporting Summary linked to this article.

## Data availability
Novel genetic variants described in this study have been deposited in ClinVar, the corresponding accession codes and permanent links are provided in Supplementary Data 4. Previously published variants in FKBP6 are available in ClinVar under accession numbers SCV002507290 [https://www.ncbi.nlm.nih.gov/clinvar/variation/1684032], SCV002507292 [https://www.ncbi.nlm.nih.gov/clinvar/variation/1684033], and SCV002507294 [https://www.ncbi.nlm.nih.gov/clinvar/variation/1684034]. Submission of human exome/genome sequencing data from the MERGE cohort, the Strasbourg cohort, and the Barcelona cohort to a repository is not covered by the proband's informed consent. These data will be available upon request for academic use and within the limitations of the proband's informed consent by contacting frank.tuettelmann@ukmuenster.de. Each request will be reviewed within 1 month and the researcher will need to sign a data access agreement. Sequencing data from the Nijmegen cohort and piRNA-seq data (.fastq files) have been deposited in the European Genome-phenome Archive (EGA) under restricted access under the accession codes EGAS00001005417 and EGAS50000000397. These data will be available upon request for academic use and within the limitations of the provided informed consent by applying for access through the EGA's online form. Every request will be reviewed within 4 weeks by the respective Data Access Committee and the researcher will need to sign a data access agreement after approval. Accession codes for AlphaFold2 structures are AF-Q9NQI0-F1 [https://alphafold.ebi.ac.uk/entry/Q9NQI0] for DDX4, AF-Q8WW33-F1 [https://alphafold.ebi.ac.uk/entry/Q8WW33] for GTSF1, AF-Q5T8I9-F1 [https://alphafold.ebi.ac.uk/entry/Q5T8I9] for HENMT1, AF-Q9BXT6-F1 [https://alphafold.ebi.ac.uk/entry/Q9BXT6] for MOV10L1, AF-Q8TC59-F1 [https://alphafold.ebi.ac.uk/entry/Q8TC59] for PIWIL2, AF-Q9BXT4-F1 [https://alphafold.ebi.ac.uk/entry/Q9BXT4] for TDRD1, AF-Q8NDG6-F1 [https://alphafold.ebi.ac.uk/entry/Q8NDG6] for TDRD9, and AF-Q587J7-F1 [https://alphafold.ebi.ac.uk/entry/Q587J7] for TDRD12. For GPAT2 (NM_001321526.1), the.pdb file is provided as Supplementary Data 2. For PNLDC1 (NM_001271862.2), the.pdb file is provided as Supplementary Data 3. Source data are provided with this paper.

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

## Acknowledgements
The authors gratefully thank all patients and their family members for providing data and samples and their consent for genomic analyses. The CeRA's physicians are thanked for taking care of the patients and documenting all clinical data in the database Androbase©. Moreover, the authors thank Christina Burhöi for her excellent technical support. F.T. was supported by the Interdisciplinary Centre for Clinical Research Münster (IZKF, Tüt4/011/23). F.T., C.F., and N.N. were supported by the Deutsche Forschungsgemeinschaft (DFG, German Research Foundation) sponsored Clinical Research Unit 'Male Germ Cells' (CRU326, project number 329621271). F.T., J.S., N.N., and S.K. were supported by the German Federal Ministry for Education and Research (BMBF) as part of the Junior Scientist Research Centre "ReproTrack.MS" (grant 01GR2303). C.B. was supported by a grant of the MedK program of the Medical Faculty Münster. C.K. and A.R.E. were funded by the Spanish Ministry of Health Instituto Carlos III-FIS (grant numbers PI17/01822 and PI20/01562). J.A.V. was supported by an Investigator Award in Science from the Wellcome Trust (209451). The authors acknowledge the use of Biorender that was used to create parts of schematic Fig. 1, Fig. 5, and Supplementary Fig. 1.

## Author contributions
B.S. and F.T. conceived and designed the experiments and wrote the manuscript. B.S., D.F.C., S.V., and C.K. supervised the experiments. C.B., R.S., A.-K.D., F.G., L.M., J.S., Ö.O., N.L. performed the experiments. A.Z., D.M., A.R.E., M.J.X., C.R., M.J.W., R.S., S.D. analyzed the data. S.K., D.F., A.P., G.V., C.K., J.A.V., K.M., A.S.G., and A.S. contributed patient samples. A.Z., C.F., M.J.W., G.V., N.N., J.A.V., S.V., D.F.C., and D.O.C. provided critical feedback and helped shape the research and manuscript. All authors revised and approved the final version of the manuscript.

## Funding

## Competing interests
The authors declare no competing interests.

## Additional information

[1]Centre of Medical Genetics, Institute of Reproductive Genetics, University of Münster, Münster, Germany. [2]Division of Genetics, Oregon National Primate Research Center, Oregon Health & Science University, Portland, OR, USA. [3]Laboratory of Molecular Neurooncology, Neuroscience Institute, Lithuanian University of Health Sciences, Kaunas, Lithuania. [4]Centre for Regenerative Medicine, Institute for Stem Cell Research, School of Biological Sciences,

University of Edinburgh, Edinburgh, UK. [5]Wellcome Centre for Cell Biology, School of Biological Sciences, The University of Edinburgh, Edinburgh, UK. [6]Laboratoire de Génétique Médicale LGM, institut de génétique médicale d'Alsace IGMA, INSERM UMR 1112, Université de Strasbourg, Strasbourg, France. [7]Hôpital Universitaire de Bruxelles, Hôpital Erasme, Service de Gynécologie-Obstétrique, Clinique de Fertilité, Université libre de Bruxelles (ULB), Bruxelles, Belgium. [8]Institute of Veterinary Anatomy, Histology and Embryology, Justus-Liebig-Universität Gießen, Gießen, Germany. [9]Clinic for Urology, Paediatric Urology and Andrology, Justus Liebig University Gießen, Gießen, Germany. [10]Andrology Department, Fundació Puigvert, Universitat Autònoma de Barcelona, Instituto de Investigaciones Biomédicas Sant Pau, Barcelona, Catalonia, Spain. [11]Biosciences Institute, Faculty of Medical Sciences, Newcastle University, Newcastle upon Tyne, UK. [12]Centre of Medical Genetics, Department of Medical Genetics, University of Münster, Münster, Germany. [13]Centre of Reproductive Medicine and Andrology, Department of Clinical and Surgical Andrology, University Hospital Münster, Münster, Germany. [14]Department of Gynecology and Obstetrics Novafertil IVF Center, Konya, Turkey. [15]Department of Andrology Novafertil IVF Center, Konya, Turkey. [16]Newcastle Fertility Centre, The Newcastle upon Tyne Hospitals NHS Foundation Trust, Newcastle upon Tyne, UK. [17]Department of Obstetrics and Gynecology, Radboud University Medical Center, Nijmegen, The Netherlands. [18]Department of Experimental and Clinical Biomedical Sciences "Mario Serio", University of Florence, University Hospital Careggi, Florence, Italy. [19]Laboratoire de Diagnostic Génétique, UF3472-génétique de l'infertilité, Hôpitaux Universitaires de Strasbourg, Strasbourg, France. ✉e-mail: frank.tuettelmann@ukmuenster.de

