## [Peer Review File · Nature Communications]

REVIEWER COMMENTS

Reviewer #1 (Remarks to the Author):

Mouse genetic studies have revealed the importance of the piRNA pathway in mammalian spermatogenesis and male fertility. However, its direct relevance to human fertility remains poorly understood. In this manuscript, by analyzing 4 cohorts of more than 2000 infertile men, the authors describe a large collection of piRNA disfunctions in 39 infertile men with biallelic variants in 14 different piRNA pathway genes each with varying phenotypic severity. They draw comparison with current mouse models studying the same genes. Furthermore, they identified co-dependences between these mutated genes with other proteins in the piRNA pathway in humans. Overall, the authors show that these mutated piRNA pathway genes found in human patients lead to detrimental effects in piRNA functions and male fertility. This represents the most comprehensive report of piRNA pathway gene mutations in human infertility thus far, which is of great importance to linking piRNA biology to human reproduction.

Major Points:

- The authors describe each mutation well, but do not draw much direct comparison to known mouse models of the same effected gene although they claim differences are seen between species except briefly in the discussion section. It may be helpful to show a clear comparison between human and mouse phenotypes in the main text. Figure 6B does this well.
- Fig. 3c: The authors suggest that GPAT2 expression is strikingly reduced (still exist) in M2556. The mutation of patient M2556 is p.(Glu386Valfs*16), meaning that GPAT2 translation is stopped at the 402 amino acid. The immunogen of GPAT2 antibody used in this study (Sigma Aldrich, HPA036841) is amino acid residues 474-546 of human GPAT2. This antibody definitely cannot detect the truncated GPAT2 in M2556 testis. Please re-evaluate if the GPAT2 signal is real or false positive.
- Given that many human cases shown here have not been extensively proven to be protein null, the germ cell phenotypes are not directly comparable with corresponding KO mice. The author should be cautious in drawing the conclusion as in line 59 and elsewhere “testicular phenotypes repeatedly differ from those of the respective knockout mice”.
- For the differences in phenotype between humans and mice, the authors do not investigate this further than stating the phenotypes. It would be helpful for the authors to provide their thoughts as to why differences are observed between species with the same gene and pathway if they want to emphasize that.
- Line 315: homozygous LoF variants in PIWIL1 have no LINE1 ORFp1 expression. This differs from Piwil1 KO mice which have significant LINE1 up-regulation. The authors should briefly discuss this in Discussion.

- Figure 6b: PnlDC1 KO mice have been reported as elongated spermatid arrest with sporadic meiotic arrest.

Minor Points:

- Fig. 1b, third panel, it should be “posttranscriptional” silencing.
- Fig. 4b, it would be helpful to provide a control testis histology and label genes.
- Line 799-800: for abbreviations, ES is not present in figures, while SG is not in the text. There are several similar issues about abbreviations in other figure legends.
- Supplementary Fig 1: some graphics block gene names.
- The labeling of genes and mutations and patient ID is not consistent. Consider adding patient ID in the text or adding genes and mutations in the figures.

Reviewer #2 (Remarks to the Author):

In this paper, Stallmeyer et al. characterize the genetic mutations in the piRNA pathway genes in infertile men. They identified altogether 39 infertile men carrying biallelic variants in 14 different piRNA pathway genes. These included the piRNA-binding protein PIWIL1 and PIWIL2 and the effector protein GTSF1, piRNA biogenesis factors (MOV10L1, PLD6, GPAT2, PNLDC1, HENMT1, MAEL, FKBP6, DDX4), as well as scaffolding Tudor-domain containing proteins (TDRD1, 9, 12). Various knockout mouse models for piRNA pathway components have previously revealed their importance for spermatogenesis and male fertility in mice. piRNA machinery is known to be conserved in human, and some mutations in the piRNA pathway components have been reported in infertile men. However, this is the first systematic analysis of piRNA pathway mutations in human, which clearly revealed the conservation of the critical function of piRNA pathways in human spermatogenesis, and highlights piRNA pathway as a major contributor to human spermatogenic failure.

In addition to the detailed characterization of the mutations, the author provide information about the spermatogenic phenotype of the variant carriers as well as the expression of the piRNA pathway components in their testis. Some mutant testes were analysed by small-RNAseq to show the reduced levels of piRNA cluster-mapping reads. As a demonstration of the functional consequences of the mutations, the authors also performed immunohistochemistry of LINE1 ORFp1 to reveal defective transposon silencing in some of the variant carriers. They also very nicely relate their findings in infertile men to previous studies using knockout mouse models and provide

very useful comparison of defects caused by the mutation of specific factors in human vs. mouse. Overall, this study is an important, carefully conducted study that provides critical novel information about the genetic landscape of male infertility. Furthermore, the manuscript is well-written, and the tables and figures contain detailed information that will be a highly valuable resource for future studies.

Specific comments:

Introduction contains some inaccuracies that should be revised, for example:

- The first paragraph: please include references.
- Lines 81-82: even though pre-pachytene piRNAs are named as pre-pachytene because they are expressed at pre-pachytene stage before the pachytene piRNAs are induced, they are still found in pachytene spermatocytes and round spermatids (and actually can also be found associated with PIWIL1). Therefore, I suggest to revise the sentence and avoid stating too strongly about restricted expression pattern and exclusive association with PIWIL2 (e.g. ... pre-pachytene piRNAs that are already expressed in early spermatogenic cells before the pachytene phase, are mainly associated with PIWIL2).
- Line 86, “transcriptional degradation”, not sure what do you mean by this, please modify
- Line 87, what do you mean by “at the late and post-meiotic stages of spermatogenesis, some words missing/too much?”

Lines 282-294: Because of the severe disruption of spermatogenesis in the mutant carriers, it is very difficult to conclude about the level of expression of the piRNA pathway proteins. For example, PIWIL1 is expressed quite late in spermatogenesis (late pachytene spermatocyte and round spermatids), and many studied variant carriers in Fig. 5a appear to have spermatogenesis arrested before these phases. Therefore, the reduced expression may be because of the lack of PIWIL1-expressing cells. Also, without proper quantifiable experiments, it is not appropriate to claim too strongly about the changes in the expression levels. Please modify the paragraph to take this into consideration.

Lines 302-306: The changes in the size distribution of piRNAs in variant carriers most probably reflect the spermatogenic arrest. The longer PIWIL1-bound piRNA (28-31 nt) are expressed later during spermatogenesis, in late pachytene spermatocytes and round spermatids, and the reduced amount of these longer piRNAs is can be caused by the absence/greatly reduced number of these cells, leading to relative higher expression of PIWIL2-bound shorter piRNAs.

Lines 307-316: It is surprising to see the transposon expression in spermatogonia, not spermatocytes. Immunohistochemistry is difficult to control due to the lack of ORFp1 expression in normal testis, and therefore, it is challenging to judge the performance of the antibody. It would be much more convincing to show the induced expression of ORFp1 by western blotting. Are there any frozen testicular material available from any the variant carriers for this kind of experiment? The validation would be important considering the big statement made already in the title and the abstract about the transposon de-repression.

Lines 330-333: It is very difficult to get the point of this sentence, please clarify.

Line 578: It would be better to use piRNA clusters identified on the basis of piRNA precursor transcripts to map reads to the piRNA clusters (PMID: 31900453)

Minor comments

- Line 773: biogenesis-relate -> biogenesis-related

Reviewer #3 (Remarks to the Author):

Germline small RNAs called piRNAs are tasked with silencing of transposable elements in the animal germline genome. This is a conserved process in organisms ranging from flies to mice and human. Study of mouse mutants for components in the piRNA pathway is the only available resource for researchers in understanding the relevance of small RNAs in ensuring genome integrity. Recently, study of hamster mutants for some of the components have complemented these studies, and also pointed out that mice may not be very representative of how piRNAs operate in the mammalian germline. For example, the piRNA pathway is not essential for transposon silencing and fertility in female mice, but is critical for female hamsters. Sequencing of cow, monkey and human oocytes have previously indicated the expression of piRNAs. It is in this context, this current study is so valuable.

The authors describe their identification of human mutations in the piRNA pathway components that result in male infertility. First, they identify 14 key human piRNA pathway genes with mutations that result in male fertility defects (this study focused on male fertility only). Second, They find that phenotypes noted in mouse mutants differ from that found in the human patients. Third, there is a certain dependency between loss of a particular factor and expression of the other piRNA factors, which is interesting. I wonder if this is due to germ cell loss or actually due to a causal relationship between a particular factor and transcription of the other genes/or their stability or translation. Fourth, they link the presence of the sterility mutations to activation of transposable elements in

humans, which is not extensively studied so far. Fifth, they demonstrate an impact on piRNA biogenesis and accumulation in several of the patients.

Overall, it is a valuable study that will be a reference material for labs studying the mechanistic basis of piRNA biogenesis, especially the knowledge of specific human mutations affecting function will be useful. In this context, the clear figures with mutations placed on to the structure predictions allow for a better understanding of potential consequences. I support is publication.

Point by point response to the reviewers' comments

Title: Inherited defects of piRNA biogenesis cause transposon de-repression, impaired spermatogenesis, and human male infertility

Reviewer #1 (Remarks to the Author)

Mouse genetic studies have revealed the importance of the piRNA pathway in mammalian spermatogenesis and male fertility. However, its direct relevance to human fertility remains poorly understood. In this manuscript, by analyzing 4 cohorts of more than 2000 infertile men, the authors describe a large collection of piRNA disfunctions in 39 infertile men with biallelic variants in 14 different piRNA pathway genes each with varying phenotypic severity. They draw comparison with current mouse models studying the same genes. Furthermore, they identified co-dependences between these mutated genes with other proteins in the piRNA pathway in humans. Overall, the authors show that these mutated piRNA pathway genes found in human patients lead to detrimental effects in piRNA functions and male fertility. This represents the most comprehensive report of piRNA pathway gene mutations in human infertility thus far, which is of great importance to linking piRNA biology to human reproduction.

Dear reviewer,

thank you very much for your positive feedback on our manuscript. Please find below our responses to your very valuable comments, which definitely helped to further improve the quality of our study.

Major Points:

- The authors describe each mutation well, but do not draw much direct comparison to known mouse models of the same effected gene although they claim differences are seen between species except briefly in the discussion section. It may be helpful to show a clear comparison between human and mouse phenotypes in the main text. Figure 6B does this well.

We agree that the comparison of the reproductive phenotypes between human variant carriers and mouse knockout models of the respective genes is of importance and have therefore added a new section in the Results that provides an overview on this topic (lines 343ff). Also as suggested, we have moved the former Figure 6b to the Results section (now Figure 6a).

- Fig. 3c: The authors suggest that GPAT2 expression is strikingly reduced (still exist) in M2556. The mutation of patient M2556 is p.(Glu386Valfs*16), meaning that GPAT2 translation is stopped at the 402 amino acid. The immunogen of GPAT2 antibody used in this study (Sigma Aldrich, HPA036841) is amino acid residues 474-546 of human GPAT2. This antibody definitely cannot detect the truncated GPAT2 in M2556 testis. Please re-evaluate if the GPAT2 signal is real or false positive.

Thank you very much for highlighting this discrepancy. We agree that the GPAT2 antibody will not detect a truncated protein that might be translated from a *GPAT2* mRNA of M2556, who is carrier of the homozygous splice acceptor site variant c.1156-1G>A that results in skipping of exon 12.

Therefore, we repeated the staining and tried different blocking conditions. By diluting the antibody in blocking solution instead of BSA and increasing the blocking time, the faint staining in spermatocytes of M2556 was no longer evident, while spermatocytes of control testicular sections, as well as several of the other piRNA factor samples, revealed a clear cytoplasmic staining in spermatocytes. We have included these updated data in Figure 3c and Supplementary Figures 12 and 15b. Once more, thank you for spotting this so that we could resolve this issue.

Given that many human cases shown here have not been extensively proven to be protein null, the germ cell phenotypes are not directly comparable with corresponding KO mice. The author should be cautious in drawing the conclusion as in line 59 and elsewhere “testicular phenotypes repeatedly differ from those of the respective knockout mice”.

This has indeed been a point we have been discussing intensively while preparing the manuscript. We agree that more human biallelic variant carriers for each of the genes need to be identified and phenotypically analyzed in order to draw firm conclusions about differences or similarities in reproductive phenotypes between mice and men and have changed the wording in line 59 accordingly. In addition, we also address these limitations of the study in the Discussion (lines 393ff).

- For the differences in phenotype between humans and mice, the authors do not investigate this further than stating the phenotypes. It would be helpful for the authors to provide their thoughts as to why differences are observed between species with the same gene and pathway if they want to emphasize that.

We expanded the discussion to address this very good suggestion (lines 447ff).

- Line 315: homozygous LoF variants in PIWIL1 have no LINE1 ORFp1 expression. This differs from Piwil1 KO mice which have significant LINE1 up-regulation. The authors should briefly discuss this in Discussion.

We have also expanded the section in the Discussion where we discuss these observed differences in the LINE1 upregulation. (lines 430ff).

- Figure 6b: Pnlcd1 KO mice have been reported as elongated spermatid arrest with sporadic meiotic arrest.

Thank you for spotting this inaccuracy. We have corrected Figure 6a accordingly and also highlight the genes associated with postmeiotic arrest in the Results.

Minor Points:

- Fig. 1b, third panel, it should be “posttranscriptional” silencing.

We corrected Figure 1b accordingly.

- Fig. 4b, it would be helpful to provide a control testis histology and label genes.

Thank you for this valuable suggestion. We have included a histology of testis with full spermatogenesis. In each figure, we now indicate the patient ID, the identified gene and variant, and the antibody used for IHC.

- Line799-800: for abbreviations, ES is not present in figures, while SG is not in the text. There are several similar issues about abbreviations in other figure legends.

Thank you for spotting these inconsistencies! We have checked all figure legends and corrected the abbreviations accordingly.

- Supplementary Fig 1: some graphics block gene names.

We have corrected this and all gene names are now visible.

- The labeling of genes and mutations and patient ID is not consistent. Consider adding patient ID in the text or adding genes and mutations in the figures.

As suggested, we have changed the labeling in all figures to include the patient IDs, identified variant, and antibody used for IHC. In addition, we have included the patient IDs in the main text when describing each of the identified variants.

Reviewer #2 (Remarks to the Author)

In this paper, Stallmeyer et al. characterize the genetic mutations in the piRNA pathway genes in infertile men. They identified altogether 39 infertile men carrying biallelic variants in 14 different piRNA pathway genes. These included the piRNA-binding protein PIWIL1 and PIWIL2 and the effector protein GTSF1, piRNA biogenesis factors (MOV10L1, PLD6, GPAT2, PNLDC1, HENMT1, MAEL, FKBP6, DDX4), as well as scaffolding Tudor-domain containing proteins (TDRD1, 9, 12). Various knockout mouse models for piRNA pathway components have previously revealed their importance for spermatogenesis and male fertility in mice. piRNA machinery is known to be conserved in human, and some mutations in the piRNA pathway components have been reported in infertile men. However, this is the first systematic analysis of piRNA pathway mutations in human, which clearly revealed the conservation of the critical function of piRNA pathways in human spermatogenesis, and highlights piRNA pathway as a major contributor to human spermatogenic failure.

In addition to the detailed characterization of the mutations, the author provide information about the spermatogenic phenotype of the variant carriers as well as the expression of the piRNA pathway components in their testis. Some mutant testes were analysed by small-RNAseq to show the reduced levels of piRNA cluster-mapping reads. As a demonstration of the functional consequences of the mutations, the authors also performed immunohistochemistry of LINE1 ORFp1 to reveal defective transposon silencing in some of the variant carriers. They also very nicely relate their findings in infertile men to previous studies using knockout mouse models and provide very useful comparison of defects caused by the mutation of specific factors in human vs. mouse. Overall, this study is an important, carefully conducted study that provides critical novel information about the genetic landscape of male infertility. Furthermore, the manuscript is well-written, and the tables and figures contain detailed information that will be a highly valuable resource for future studies.

Dear reviewer,

Thank you very much for the very positive evaluation of our study. We are delighted that you share our opinion that this manuscript is of critical relevance for the broad audience of Nature Communications. Please find below our answers to your suggestions that helped to improve the overall quality of the manuscript.

Introduction contains some inaccuracies that should be revised, for example:

- The first paragraph: please include references.

- Lines 81-82: even though pre-pachytene piRNAs are named as pre-pachytene because they are expressed at pre-pachytene stage before the pachytene piRNAs are induced, they are still found in pachytene spermatocytes and round spermatids (and actually can also be found associated with PIWIL1). Therefore, I suggest to revise the sentence and avoid stating too strongly about restricted expression pattern and exclusive association with PIWIL2 (e.g. ... pre-pachytene piRNAs that are already expressed in early spermatogenic cells before the pachytene phase, are mainly associated with PIWIL2).

Thank you very much for spotting these inaccuracies. We have revised the sentences in question accordingly (lines 84ff) and included references.

- Line 86, “transcriptional degradation”, not sure what do you mean by this, please modify

We have modified this sentence to clarify (lines 89ff)

- Line 87, what do you mean by “at the late and post-meiotic stages of spermatogenesis, some words missing/too much?”

Thank you for identifying this mistake. We have corrected the sentence (lines 89ff).

Lines 282-294: Because of the severe disruption of spermatogenesis in the mutant carriers, it is very difficult to conclude about the level of expression of the piRNA pathway proteins. For example, PIWIL1 is expressed quite late in spermatogenesis (late pachytene spermatocyte and round spermatids), and many studied variant carriers in Fig. 5a appear to have spermatogenesis arrested before these phases. Therefore, the reduced expression may be because of the lack of PIWIL1-expressing cells. Also, without proper quantifiable experiments, it is not appropriate to claim too strongly about the changes in the expression levels. Please modify the paragraph to take this into consideration.

Thank you for raising this point. We agree that the data presented on the expression of other proteins of the piRNA pathway are rather preliminary and should not be over-interpreted. Therefore, we have reworded the conclusion in the corresponding paragraph (line 313) and additionally point out the limitations of these analyses in the Discussion (lines 413ff).

Lines 302-306: The changes in the size distribution of piRNAs in variant carriers most probably reflect the spermatogenic arrest. The longer PIWIL1-bound piRNA (28-31 nt) are expressed later during spermatogenesis, in late pachytene spermatocytes and round spermatids, and the reduced amount of these longer piRNAs is can be caused by the absence/greatly reduced number of these cells, leading to relative higher expression of PIWIL2-bound shorter piRNAs.

Thank you very much for this valuable comment. As the number of samples included in the piRNA-seq analysis is indeed small, we agree that we cannot exclude that the observed differences in piRNA length are a direct consequence of the varying testicular phenotypes and,

thus, germ cell compositions in the seminiferous tubules. Therefore, we have deleted the respective figure and the comment from the results.

Lines 307-316: It is surprising to see the transposon expression in spermatogonia, not spermatocytes. Immunohistochemistry is difficult to control due to the lack of ORFp1 expression in normal testis, and therefore, it is challenging to judge the performance of the antibody. It would be much more convincing to show the induced expression of ORFp1 by western blotting. Are there any frozen testicular material available from any the variant carriers for this kind of experiment? The validation would be important considering the big statement made already in the title and the abstract about the transposon de-repression.

Thank you very much for highlighting this topic because it is indeed both surprising and important. Frozen testicular tissue from the respective variant carriers is in general only rarely available, because the biopsies are obtained from testicular sperm extraction attempts with the ultimate aim of assisted reproduction via intracytoplasmic sperm injection and only the material left after this procedure can be stored for research purposes.

However, because this is such an important point, we have performed IHC staining with another monoclonal antibody raised against LINE1 ORF1p. The staining results are shown in Figure S16b,c and fully confirm the previous results concerning expression in spermatogonia. Interestingly, with this new monoclonal antibody, we observed LINE1 ORF1p-positive germ cells in single tubules of the *TDRD9* variant carrier M800, who was negative in the initial staining. Therefore, we repeated the staining with both LINE1 OFR1p antibodies and confidently confirmed the positive staining result in selected tubules (Supplementary Figure 16b,c).

Both LINE1 ORF1p antibodies were obtained from the same company (Abcam), but the supplier stated on request that they were derived from independent clones. We never observed any LINE1 ORF-1p positive staining in any of the control testicular tissues analyzed. In contrast we observed a concordant positive staining result in all *TDRD12* (3 patients), *FKBP6* (3 patients) and *GPAT2* (2 patients), which strongly supports the validity of the results.

Lines 330-333: It is very difficult to get the point of this sentence, please clarify.

Here we wanted to discuss on a publication that previously linked heterozygous missense variants in *PIWIL1* to azoospermia (PMID: 28552346), assuming an autosomal dominant inheritance. The genetic data presented in this publication were critically discussed in a comment (PMID: 33861957) and our finding of a homozygous LoF variant in *PIWIL1* underlines that *PIWIL1* is most likely also a recessive disease gene. We have moved and revised this sentence to clarify this point (lines 372ff).

Line 578: It would be better to use piRNA clusters identified on the basis of piRNA precursor transcripts to map reads to the piRNA clusters (PMID: 31900453)

We fully understand your suggestion. We had decided to map the small RNAseq reads based on the piRNA-producing loci defined by Girard *et al.* (Nature, 2006; PMID: 16751776) because this analysis has already been used in a recent publication on the impact of human variants in *FKBP6* on biogenesis of pachytene piRNAs, and we wanted to compare our results with the results of that study. Indeed, Özata *et al.* showed that there are more piRNAs mapping to their suggested piRNA-producing loci compared to other suggested loci. However, the reported difference in the percentage of mapped piRNAs between Özata *et al.* and Girard *et al.* piRNA-producing loci is actually rather small (see black and green lines in the Extended Data Fig. 4 of PMID: 31900453):

[Redacted]

Please refer to Extended Data Figure 4 of Özata, D.M., Yu, T., Mou, H. et al. Evolutionarily conserved pachytene piRNA loci are highly divergent among modern humans. *Nat Ecol Evol* 4, 156–168 (2020). <https://doi.org/10.1038/s41559-019-1065-1>

Thus, our results would very likely not significantly differ depending on the basis for mapping. Further, we now use the opportunity to compare our piRNA sequencing results with those identified in *FKBP6* variant carriers (PMID: 36150389), which was only possible by sticking to the same mapping. We have included this comparison in a new Supplementary Figure 16a.

Minor comments

- Line 773: biogenesis-relate -> biogenesis-related

We corrected the wording accordingly.

Reviewer #3 (Remarks to the Author)

Germline small RNAs called piRNAs are tasked with silencing of transposable elements in the animal germline genome. This is a conserved process in organisms ranging from flies to mice and human. Study of mouse mutants for components in the piRNA pathway is the only available resource for researchers in understanding the relevance of small RNAs in ensuring genome integrity. Recently, study of hamster mutants for some of the components have complemented these studies, and also pointed out that mice may not be very representative of how piRNAs operate in the mammalian germline. For example, the piRNA pathway is not

essential for transposon silencing and fertility in female mice, but is critical for female hamsters. Sequencing of cow, monkey and human oocytes have previously indicated the expression of piRNAs. It is in this context, this current study is so valuable.

The authors describe their identification of human mutations in the piRNA pathway components that result in male infertility. First, they identify 14 key human piRNA pathway genes with mutations that result in male fertility defects (this study focused on male fertility only). Second, They find that phenotypes noted in mouse mutants differ from that found in the human patients. Third, there is a certain dependency between loss of a particular factor and expression of the other piRNA factors, which is interesting. I wonder if this is due to germ cell loss or actually due to a causal relationship between a particular factor and transcription of the other genes/or their stability or translation. Fourth, they link the presence of the sterility mutations to activation of transposable elements in humans, which is not extensively studied so far. Fifth, they demonstrate an impact on piRNA biogenesis and accumulation in several of the patients.

Overall, it is a valuable study that will be a reference material for labs studying the mechanistic basis of piRNA biogenesis, especially the knowledge of specific human mutations affecting function will be useful. In this context, the clear figures with mutations placed on to the structure predictions allow for a better understanding of potential consequences. I support is publication.

Dear reviewer,

Thank you very much for the very positive evaluation of our study. We agree that this manuscript will be a valuable tool for labs studying the mechanistic basis of piRNA biogenesis and will be of broad interest not only to researchers who are working on genetic causes of male infertility, but also to those interested in the functional characterization of proteins involved in piRNA biogenesis and piRNA-mediated functions.

REVIEWERS' COMMENTS

Reviewer #1 (Remarks to the Author):

The authors have addressed all my comments and concerns. Congratulations on a significant contribution to the piRNA field.

Reviewer #2 (Remarks to the Author):

The authors have addressed all my comments and modified the manuscript accordingly. I do not have further concerns.